# SOUNDCAM: A Dataset for Finding Humans Using Room Acoustics

**Mason Wang**[*1]    **Samuel Clarke**[*1]
**Jui-Hsien Wang**[2]    **Ruohan Gao**[1]    **Jiajun Wu**[1]

[1]Stanford University        [2]Adobe Research

## Abstract

A room's acoustic properties are a product of the room's geometry, the objects within the room, and their specific positions. A room's acoustic properties can be characterized by its impulse response (RIR) between a source and listener location, or roughly inferred from recordings of natural signals present in the room. Variations in the positions of objects in a room can effect measurable changes in the room's acoustic properties, as characterized by the RIR. Existing datasets of RIRs either do not systematically vary positions of objects in an environment, or they consist of only *simulated* RIRs. We present SOUNDCAM, the largest dataset of unique RIRs from in-the-wild rooms publicly released to date.[1] It includes 5,000 10-channel real-world measurements of room impulse responses and 2,000 10-channel recordings of music in three different rooms, including a controlled acoustic lab, an in-the-wild living room, and a conference room, with different humans in positions throughout each room. We show that these measurements can be used for interesting tasks, such as detecting and identifying humans, and tracking their positions.

## 1   Introduction

The physical sound field of a room, or the transmission and reflection of sound between any two points within the room, is influenced by many factors, including the geometry of the room as well as the shape, position, and surface material of each object within the room [Kuster et al., 2004, Bistafa and Bradley, 2000, Cucharero et al., 2019]. For this reason, each room has a distinct sound field, and a change in a position of a particular object in the room will generally change this sound field in distinct ways [Mei and Mertins, 2010, Götz et al., 2021]. We present SOUNDCAM, a novel dataset to investigate whether this principle can be used for three distinct tasks for identifying humans and tracking their positions in diverse conditions, including in both controlled and in-the-wild rooms.

A room's sound field between a given source and listener location pair can be characterized by a room impulse response (RIR) between that pair. While there are many datasets of RIRs recorded in diverse environments, existing datasets mainly focus on densely characterizing the sound field of rooms by varying the locations of the sources and/or listeners in the environment. There are datasets which focus on isolating the effects of changing objects or their positions in the room, but they provide RIRs generated purely in *simulated* environments. Our SOUNDCAM dataset is the largest dataset of unique, real RIRs from in-the-wild rooms publicly released to date and specifically focuses on isolating the acoustic effects of changing the positions and identities of humans.

The ability to identify and/or track the position of a human in an indoor environment has many applications to ensure safe and high-quality user experiences for interactive applications such as

---

[*]indicates equal contribution

[1]The project page and dataset are available at `https://masonlwang.com/soundcam/`, with data released to the public under the MIT license.

37th Conference on Neural Information Processing Systems (NeurIPS 2023) Track on Datasets and Benchmarks.

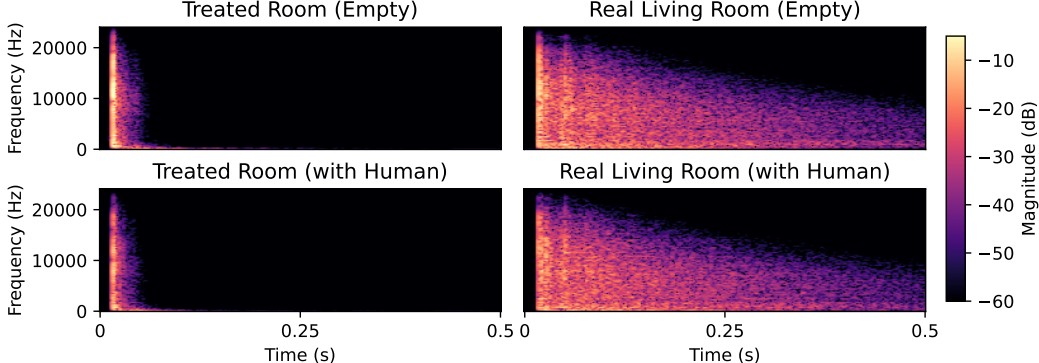

Figure 1: Spectrograms visualizing the RIRs from the Treated Room (**left column**) and real Living Room (**right column**), either empty (**top row**) or with humans standing near the loudspeaker sound source (**bottom row**). The RIRs within each column are from the same speaker and microphone position. While the human's obstructing the direct path noticeably attenuates the intensity and duration of the RIR as measured by the microphone in each room, the Living Room has much stronger *indirect* paths for the sound to reach the microphone through reflections and thus shows less obvious effects.

virtual/augmented reality and smart home assistants. While many prior works have successfully tracked humans and objects using vision signals, we instead focus on identifying and tracking targets using audio signals. Sound waves undergo diffraction, making it possible to curve around obstacles (e.g., you can hear a speaker behind a sofa playing music even when you cannot see it). Audio signals also occupy a broad frequency spectrum, making them an excellent probe for the many surfaces in our everyday environment that demonstrate frequency-dependent reflection and absorption properties. For these reasons, audio signals can provide information about an environment's state which complements that provided by visual signals, to make tracking and detection more robust to edge cases such as with transparent objects [Singh and Nagla, 2019] or occlusion [Lindell et al., 2019]. Using high-resolution vision signals can also create privacy concerns in certain use cases such as in healthcare settings [Zhang et al., 2012, Chou et al., 2018], where acoustic features could provide more privacy. Conversely, using audio signals to track and identify humans may also have applications in covert surveillance, and we claim that this makes releasing a public dataset and benchmark to the academic community essential to mitigating the risk of malicious use on an unwitting party.

We thus collect a large dataset of 5,000 10-channel RIRs from human subjects standing in varied positions across a total of three controlled and in-the-wild rooms. Each room contains data on between 2 and 5 individuals, allowing users of SOUNDCAM to check how well their methods for tracking humans generalize to unseen humans, and to develop methods for identifying humans. In real-world applications, precisely measuring RIRs using non-intrusive signals in in-the-wild rooms may be difficult. However, natural sounds are often present in real rooms, from the noise of a television, a speaker playing music, a baby crying, or a neighbor hammering nails. These natural sounds provide additional clues about the presence, identities, locations, and activities of people inside the room, even those who are not making noise. In this paper, we focus on *actively* emitted natural sounds, so SOUNDCAM also includes 2,000 10-channel recordings of music with the same annotations as the RIR measurements, in a controlled and a real-world room. This data can be used for developing methods that use more general source signals as input.

Our results show that SOUNDCAM can be used to train learning-based methods to estimate a human's location to within 30 cm of error, using room impulse responses measured by multiple microphones. However, these methods perform worse when using RIRs measured from a single microphone, when using natural music as a source signal and when the layout of the room changes. When the source music is unknown to the model and even all 10 microphones are used, our best baseline model is able to detect the presence of a human in a room with only 67% accuracy using music. Our best baseline model for identification is able to correctly classify the human from a group of five 82% of the time, when using all 10 microphones. Since placing 10 microphones in a room is less realistic for real-world applications, our results also indicate that ablating the number of microphones to 4, 2, or 1 generally reduces performance across tasks.

## 2  Related Work

**Estimating environment geometry with sound.**   Numerous works have shown how audio can be used to precisely measure salient aspects of room geometry, such as the positions of walls and furniture. Many analytical approaches require no training data in order to estimate the positions of planar walls in an environment [Dokmanić et al., 2013, Antonacci et al., 2012b]. In addition, several works use room impulse responses (RIRs) to localize inanimate acoustic reflectors  [Antonacci et al., 2012a, Aprea et al., 2009, Tervo et al., 2012].  While these methods rely on some strong assumptions of the room geometry, only claiming to map out the geometry of large planar surfaces such as walls, learning-based approaches learn to estimate environment geometries without such assumptions. Purushwalkam et al. [2021] combined camera and microphone signals gathered by a robot exploring a simulated environment, using both actively emitted and passively received audio as inputs to a learning-based framework, which estimates entire floor plans from a video sequence of only a region of the floor plan. Gao et al. [2020] showed how an agent's binaural recordings of actively emitted chirps can be used to estimate depth maps of the agent's views within a simulated environment. Christensen et al. [2020] showed similar results in real environments. Both approaches learn from datasets of chirp recordings and their corresponding ground truth depth images, learning to estimate the depth of objects within the agent's line of sight. We show how acoustic information can even be used to estimate changes in the position of a human which is occluded from the line of sight of both the sound source and the microphone.

**Tracking objects and humans with multimodal signals.**   Both passively and actively emitted sounds can be used to track and identify both objects and humans. Passive methods using sounds emitted by objects and humans while they are in motion have been used to track aircraft [Lo and Ferguson, 1999], motor vehicles [Gan et al., 2019], and human speakers [Crocco et al., 2017]. Rather than passively recording the sound emitted by the target, we record the sounds actively emitted from a speaker while a human remains silent, making active methods more relevant to our work. Yang et al. [2022] used the times of arrival (TOA) of chirping signals, emitted by specialty ultrasonic speakers, jointly with camera signals to estimate the metric scale joint pose of human bodies in household environments. They assume the bodies are within the lines of sight of both the camera and the speakers and microphones in their setup. Lindell et al. [2019] relaxed the assumption that an object must be within the acoustic line of sight by using a moving array of both microphones and co-located loudspeakers in a controlled acoustically-treated environment, emitting and recording the sounds which reflect against a wall to estimate the shape of an object behind another wall which obstructs the line of sight to the object. Our dataset includes recordings from both controlled and in-the-wild environments, including in contexts when the human is occluded from the microphone and/or speaker's line of sight. Other signals, such as radio [Adib et al., 2015, Zhao et al., 2018] and WiFi [Adib and Katabi, 2013] can be used to track human motion when the line of sight is obstructed, but these methods and that of Lindell et al. [2019] require specialized hardware setups which are not common in homes. We include recordings of both sine sweeps and popular music, played and recorded by inexpensive commodity hardware similar to what is already contained within many homes due to home assistant devices or media entertainment systems.

**Large acoustic datasets.**   Many works have released large datasets of measurements of the acoustic properties of objects and environments. The Geometric-Wave Acoustic Dataset [Tang et al., 2022] consists of 2 million synthetic room impulses generated with a wave solver-based simulation of 18,900 different scenes within virtual house layouts. The authors demonstrate how the dataset can be used for data augmentation in tasks such as automated speech recognition (ASR) or source separation. Other datasets have included measurements of real room impulse responses. The voiceHome corpus [Bertin et al., 2016] features 188 8-channel impulse responses from 12 different rooms in 3 real homes, and the dEchorate dataset [Carlo et al., 2021] comprises nearly 2,000 recordings of impulse responses of the same acoustically treated environment in 11 different configurations with varying reverberation characteristics. All our measurements are real recordings, recorded in both acoustically treated and in-the-wild environments. More importantly, our dataset measures real impulse responses of rooms with humans in different annotated positions, whereas these prior works only measure rooms with inanimate objects.

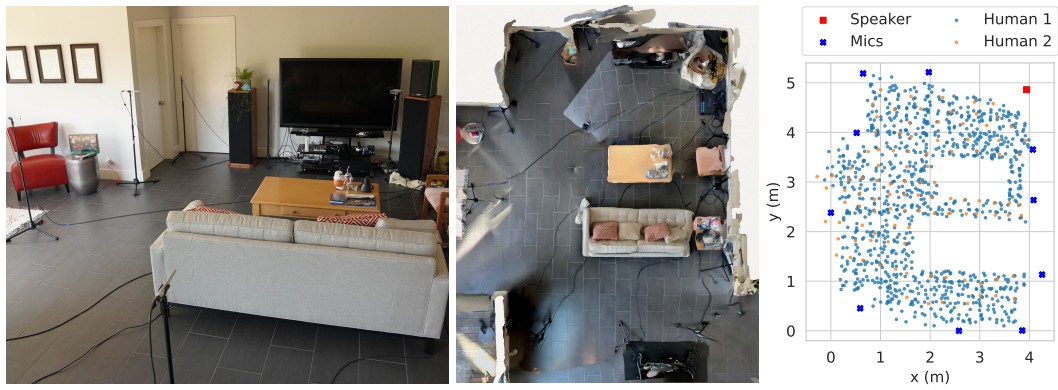

Figure 2: Images and visualizations from the real living room. (**Left**) A photo of the room. (**Middle**) An aerial view of a 3D scan of the room. (**Right**) A visualization of the microphone, speaker, and human positions in our dataset. See Appendix A for visualizations of the other rooms.

## 3  The SOUNDCAM Dataset

We collect large datasets of both sine sweeps and music clips recorded by microphones in different rooms. While we include recordings of each room while it is empty, each recording for which a human is present is paired with calibrated depth images from multiple camera angles to annotate the position of the human standing in the room while preserving subject anonymity. In total, SOUNDCAM contains multichannel sine sweep recordings from humans standing in 5,000 unique positions across three different rooms, including at least 200 RIRs from each of 5 unique humans, with an additional 2,000 recordings from humans standing in unique positions with natural music being played in a room, as summarized in Table 1. SOUNDCAM includes the largest dataset of unique RIRs from in-the-wild rooms publicly released to date. Next, we introduce our audio setup, rooms where the recording is conducted, how the human pose data is collected, and our postprocessing steps.

**Audio.**  We place a QSC K8.2 2kW Active Loudspeaker in a corner of each room, then place 10 Dayton Audio EMM6 microphones at different positions along the periphery. The 3D position of the speaker and each microphone is annotated, as shown in Figure 2. Maps of the microphone array layouts for additional rooms are included in Appendix A. The loudspeaker and microphones are connected to the same synchronized pair of MOTU 8M audio interfaces, which simultaneously play and record a signal from the loudspeaker and microphones respectively, in a time-synchronized fashion at 48kHz. Depending on the experiment, we play and record a different signal. For experiments requiring precise RIRs, we use a 10-second 20Hz-24kHz sine sweep. For experiments that test the generalization of methods to natural signals, we select a 10-second clip from the Free Music Archive [Defferrard et al., 2016], balancing the main eight genres by sampling uniformly randomly within each one. For both the sine sweeps and the music sweeps, we also record the four seconds of silence after each signal is played, to allow for the decay of the recording to be captured. The longest RT60 Reverberation Time of any of our rooms is 1.5 seconds, so the final two seconds of the recording we presume to be complete silence except for environmental and human noises.

**Rooms.**  We collect separate datasets from three rooms with diverse geometry, acoustic properties, and function. The first is an acoustically treated room without direct forced ventilation. Within the room, we collect subsets of data in each of two different static configurations: 1) with the room empty, except for recording equipment, and 2) with four fabric divider panels placed in the room, in order to test for robustness to occlusions and changes in static elements in the room. The second room is a living room in a real house (shown in Figure 2), and the third is an untreated conference room with standard furniture. We collect a textured 3D scan of each room. Additional images and floor plans of each room are included in Appendix A.

**Humans and pose.**  Our dataset includes recordings of sine sweeps and music from different humans, with the number of unique humans varying by room. We collect a 3D scan of each human, with texture and fine details omitted to preserve anonymity. We record some sine sweeps and/or music from each room while it is empty, as well as recordings where at most one human is present in each recording. For each recording with a human, the human selects a new position in the room, based on following a pattern that optimizes for even coverage of the reachable space. The human then stands straight with their arms at their sides, facing the same direction for each recording. We

| | RIR (Sine Sweep) | | Music | |
|---|---|---|---|---|
| | Humans × Recordings | Empty | Humans × Recordings | Empty |
| Treated Room | 1×1000 + 4×200 | 1000 | 1×1000 | 1000 |
| Treated Room w/ Panels | 1×1000 | 100 | 0 | 0 |
| Living Room | 1×1000 + 1×100 | 100 | 1×1000 + 1×100 | 0 |
| Conference Room | 1×1000 + 1×100 | 100 | 0 | 0 |

Table 1: Number of 10-channel recordings by room, unique human, and signal type.

| | Number of Microphones | | | |
|---|---|---|---|---|
| Treated Room | 10 | 4 | 2 | 1 |
| kNN on Levels | 54.4 (66.9) | 81.1 (57.9) | 104.3 (63.5) | 135.1 (57.4) |
| Linear Regression on Levels | 112.1 (58.8) | 130.6 (48.5) | 133.4 (47.5) | 133.8 (47.3) |
| VGGish (pretrained) | 93.6 (53.1) | 106.5 (60.2) | 97.8 (55.9) | 95.8 (55.9) |
| VGGish (multichannel) | **18.1** (13.1) | **17.2** (16.2) | **20.4** (15.4) | **71.6** (50.4) |
| Time of Arrival | 148.1 (106.2) | 133.0 (103.0) | 244.0 (117.8) | 307.6 (102.7) |
| Treated Room w/ Panels | | | | |
| kNN on Levels | 30.1 (29.0) | 61.5 (55.1) | 98.7 (72.5) | 134.6 (58.0) |
| Linear Regression on Levels | 105.7 (39.0) | 117.9 (56.7) | 136.3 (60.0) | 138.2 (51.2) |
| VGGish (pretrained) | 92.5 (60.5) | 79.9 (52.6) | 82.1 (53.5) | 104.8 (62.2) |
| VGGish (multichannel) | **13.1** (11.2) | **20.7** (19.3) | **19.6** (18.2) | **45.7** (42.2) |
| Time of Arrival | 163.7 (118.0) | 1528 (119.4) | 240.7 (121.1) | 314.3 (102.0) |
| Living Room | | | | |
| kNN on Levels | 61.6 (59.6) | 93.2 (77.1) | 123.0 (80.4) | 157.3 (60.0) |
| Linear Regression on Levels | 125.0 (62.7) | 154.5 (54.7) | 165.3 (54.2) | 168.4 (53.5) |
| VGGish (pretrained) | 142.2 (84.9) | 147.0 (86.9) | 151.0 (84.4) | 151.9 (91.3) |
| VGGish (multichannel) | **27.9** (22.0) | **23.6** (15.0) | **26.3** (21.3) | **42.0** (30.3) |
| Time of Arrival | 229.9 (150.9) | 222.5 (129.7) | 244.7 (128.5) | 308.7 (133.0) |

Table 2: Localization error of each model using sine sweep-based RIRs from varying numbers of microphones in different environments. Errors are in centimeters and "mean (stdev)" format.

position three Azure Kinect DK RGBD cameras in three corners of each room and capture RGBD images from each camera immediately before recording each sine sweep. To preserve the anonymity of our subjects, we do not release RGB images of them, and instead only release depth images and joint positions.

**Postprocessing.** In order to estimate a room impulse response (RIR) from our recordings of sine sweeps, we deconvolve the signal audio file, either sine sweep or music, from each of the microphone recordings. For annotating human pose, we use the Azure Kinect's 3D body tracking API to extract an estimate of the skeleton joint positions in the 3D coordinate frame of each camera. We then calibrate each camera's coordinate frame to the same room frame using an Aruco marker hanging in the middle of each room. We label each recording with the median of each camera's estimate of the position of the pelvis in the room coordinate frame, excluding cameras that could not successfully capture the body. See Figure 2 for a visualization of the distribution of human position annotations in the Living Room, with additional rooms shown in Appendix A. Each example's input consists of the individual simultaneous recordings from each microphone combined together in the same ordering, while the label is the estimated x-y position of the pelvis in the room coordinate frame. We release both raw recordings and postprocessed features for all our data. The number of recordings we have for each room, configuration, and unique human is summarized in Table 1.

## 4 Applications

Our dataset can be used for multiple interesting tasks in learning from acoustics, including localizing (Sec. 4.1), identifying (Sec. 4.2), and detecting (Sec. 4.3) humans. We describe the formulation, baselines, and results for each task.

| | Number of Microphones | | | |
|---|---|---|---|---|
| Treated Room | 10 | 4 | 2 | 1 |
| kNN on Levels | 133.5 (51.0) | 133.5 (48.4) | 135.8 (51.9) | 133.3 (49.2) |
| Linear Regression on Levels | 133.7 (46.9) | 133.9 (48.8) | 133.8 (47.4) | 133.7 (47.3) |
| VGGish (pre-trained) | 164.8 (79.0) | 148.0 (82.0) | 147.5 (79.4) | 155.8 (79.4) |
| VGGish (multichannel) | **31.4** (23.9) | **44.6** (34.0) | **48.6** (39.2) | **106.3** (79.6) |
| Time of Arrival | 232.4 (95.7) | 232.2 (96.9) | 219.8 (95.9) | 232.0 (97.7) |
| Living Room | | | | |
| kNN on Levels | 167.2 (56.5) | 170.0 (59.8) | 169.3 (65.7) | 168.7 (54.3) |
| Linear Regression on Levels | 125.0 (62.7) | 154.5 (54.7) | 165.3 (54.2) | 168.4 (53.5) |
| VGGish (pre-trained) | 186.6 (89.4) | 191.4 (99.9) | 199.7 (96.8) | 189.9 (100.9) |
| VGGish (multichannel) | **25.6** (18.4) | **40.3** (38.6) | **43.1** (41.6) | **82.2** (53.8) |
| Time of Arrival | 297.0 (133.0) | 298.1 (131.0) | 281.5 (127.1) | 301.7 (132.6) |

Table 3: Localization error of each model using music-based RIRs from varying numbers of microphones in different environments. Errors are in centimeters and "mean (stdev)" format.

## 4.1 Human Localization

We investigate whether different learning-based and analytical models can localize a human's position in a room from different sounds, including the RIR from a sine sweep, the RIR estimated from music signals, or a recording when the speaker is silent. In each case, we truncate to the first two seconds of each signal, as we found that the RIR had decayed by at least 60 decibels in each room within at most $\sim 1.5$ seconds. Note that this task differs from prior works which either assume the tracked object is not silent and is therefore a sound source [Gan et al., 2019, Crocco et al., 2017, Lo and Ferguson, 1999], assume that the sound source and receiver(s) are roughly collocated [Christensen et al., 2020, Gao et al., 2020, Lindell et al., 2019], or fuse estimates from both vision and ultrasonic audio [Yang et al., 2022].

**Baselines** We test both learning-based and analytical baselines for this task, as detailed below.

- kNN on Levels: We use k-Nearest Neighbors (kNN) based on the overall RMS levels from each microphone in the input, summarizing each signal by a scalar per microphone.

- Linear Regression on Levels: This is another simple learning-based baseline similar to kNN on levels except that we use Linear Regression models on the same summarized inputs.

- VGGish (pre-trained): We use the popular VGGish architecture [Hershey et al., 2017] with weights pre-trained on the AudioSet classification task [Gemmeke et al., 2017]. This pre-trained model requires reducing signals into a single channel by averaging and downsampling signals to a sample rate of 16kHz to use as input. We use the outputs of the final hidden layer as inputs to three additional layers, with a final linear layer estimating normalized coordinates.

- VGGish (multichannel): This baseline is the same as the previous one except that we use a VGGish model that preserves the channels at full 48kHz sample rate and shares the same weights for each channel. We combine the hidden features across all channels with three additional layers similar to the pre-trained version and train this entire model from scratch.

- Time of Arrival: We use an analytical method based on time of arrival (TOA), similar to that described in [Yang et al., 2022]. For each input RIR, we subtract an RIR of the empty room which is the mean of multiple trials, then use the peaks in the absolute difference to find intersecting ellipsoids based on loudspeaker and microphone positions.

Additional details and hyperparameters of each baseline are included in Appendix B.

**Results** We first test each baseline on the RIRs derived from sine sweep recordings in each room. To test the influence of using multiple microphones, we also ablate the number of microphones/channels of each RIR from 10 to 4, 2, and 1. The resulting localization errors are shown in Table 2. For all our results, we define error as the average distance in centimeters between the ground truth location and the predicted location, across the test set. We perform the same experiments for each baseline on

| | Number of Microphones | | | |
|---|---|---|---|---|
| | 10 | 4 | 2 | 1 |
| kNN on Levels | 163.6 (78.8) | 166.7 (82.8) | 164.2 (79.5) | 164.9 (85.4) |
| Linear Regression on Levels | 148.9 (59.6) | 147.2 (59.7) | 147.1 (59.8) | 147.0 (59.9) |
| VGGish (pre-trained) | 202.9 (96.9) | 195.5 (94.8) | 187.0 (96.7) | 205.7 (100.9) |
| VGGish (multichannel) | **50.7** (39.7) | **70.5** (57.0) | **67.1** (67.2) | **118.9** (74.2) |
| Time of Arrival | 304.5 (111.4) | 301.2 (108.5) | 282.8 (113.3) | 305.2 (112.3) |

Table 4: Localization error of models using music-derived RIRs in the Living Room, trained on data from one human and tested on data from another. Errors are in cm and "mean (stdev)" format.

RIRs derived from the music subset we have collected in the Treated Room and the Living Room, with resulting localization errors shown in Table 3. In both cases, the analytical method performs quite poorly, but performs especially poorly on the music-derived RIR. Deconvolving a natural music signal to derive the RIR is rather ill-posed versus deconvolving a monotonic sine sweep, so there are more likely to be spurious peaks in the RIR in the time domain, which is used by the analytical method. Further work can develop methods to estimate room acoustics given natural music signals in a way that is useful for the localization task. The poor performance of linear regression shows that the function relating the volume levels of the recording to the location is non-linear. The kNN baseline can better characterize a nonlinear landscape and thus performs better. When using more than two microphones, kNN on levels also rather consistently outperformes pre-trained VGGish, which condenses all channels to one through a mean operation before passing them through the CNN. While this suggests that the separate channels have complementary information, the improvement in the performance of pre-trained VGGish relative to kNN with one or two microphones suggests that there is important information in the spectrograms of the RIRs which is discarded when only using their levels. The multichannel VGGish takes the best of both approaches, using all channels and the full spectrograms, and consistently outperforms all models on this task. Furthermore, its performance relative to the pre-trained version on single-channel RIRs suggests that it is important to use the full 48 kHz sample rate rather than downsampling to 16 kHz.

**Generalization** To broaden the scope of potential applications of this task, we investigate the ability of our models to generalize to data outside of the training distribution. For these tasks, we only test the learning-based baselines, since the analytical time of arrival method requires no training data.

First, we use models trained on data from one human in a particular room and test generalizations to one or more other humans in the same room. Results from training models on music-derived RIRs from one human and testing on music-derived RIRs from another human in the Living Room are shown in Table 4. Though performance drops relative to training and testing on the same human, the multichannel VGGish is still able to make somewhat accurate predictions. Future work should test generalization on a more comprehensive distribution of humans of different shapes, sizes, ages, etc.

Next, we test the generalization between different layouts of furniture in the same room. Though SOUNDCAM could theoretically be used to benchmark generalization among different rooms, each room varies significantly in structure and therefore requires using a unique layout of microphones and speakers to record data, so this task is not well-defined for our baselines. On the other hand, for the Treated Room, our recordings with and without the panels both used the same placement and arrangement of speakers and microphones, with only the static contents of the room varying between recordings. We thus train each baseline model on the sine sweep RIRs from the Treated Room and test on RIRs from the Treated Room with the fabric divider panels in it, and vice versa. None of the baselines are particularly robust to the introduction of these fabric panels, with each model's mean error being at least a meter for each amount of microphones. This unsolved issue of generalization to minor changes in layout in the room could severely restrict the downstream practical applications of these models. Detailed results are included in Appendix C.1.

## 4.2 Human Identification

We also investigate whether learning-based methods can correctly classify RIRs by the identity of the human in the room. We use the RIRs from our Treated Room, where we collect at least 200 RIRs for each of the five humans. A visualization of the unique positions of each human in the room is shown at the right of Figure 3. Further, to determine whether the RIR is indeed useful

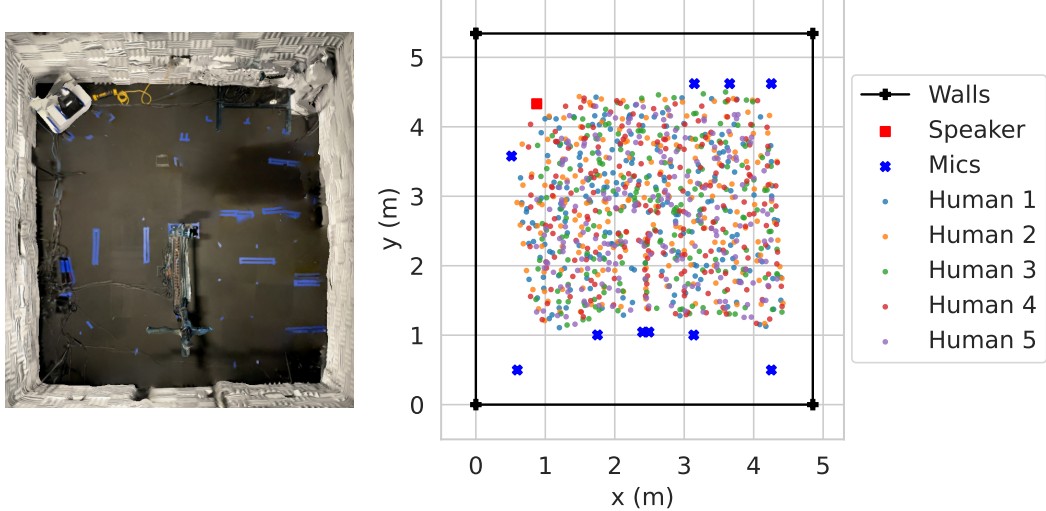

Figure 3: (**Left**) An aerial view of a 3D scan of the Treated Room. (**Right**) A visualization of the microphone, speaker, and unique human positions within the Treated Room for our human identification subset.

| Sine Sweep RIR | Number of Microphones | | | |
|---|---|---|---|---|
| | 10 | 4 | 2 | 1 |
| kNN on Levels | 65 | 46 | 42 | 28 |
| VGGish (pretrained) | 72 | 60 | 39 | 31 |
| VGGish (multichannel) | **82** | **81** | **76** | **64** |
| Silence | | | | |
| kNN on Levels | **39** | 20 | 17 | 20 |
| VGGish (pretrained) | 20 | 20 | 20 | 20 |
| VGGish (multichannel) | 20 | 20 | 20 | 20 |

Table 5: Classification accuracy (%) in identifying among five humans from both sine sweep RIRs and silence in the Treated Room.

| Music RIR | Number of Microphones | | | |
|---|---|---|---|---|
| | 10 | 4 | 2 | 1 |
| kNN on Levels | 57 | 58 | 56 | 54 |
| VGGish (pretrained) | 65 | 62 | 70 | 80 |
| VGGish (multichannel) | **99** | **100** | **100** | **98** |
| Music Raw | | | | |
| kNN on Levels | 51 | 55 | 49 | 51 |
| VGGish (pretrained) | 53 | 53 | 51 | 55 |
| VGGish (multichannel) | **75** | **77** | **71** | 55 |
| Silence | | | | |
| kNN on Levels | **59** | **54** | 54 | 52 |
| VGGish (pretrained) | 55 | 47 | 54 | **53** |
| VGGish (multichannel) | 50 | 50 | 50 | 50 |

Table 6: Classification accuracy (%) in detecting the presence of a human in the Treated Room using either a music-derived RIR, a raw recording of music, or a recording of silence.

for identification, or if the human can be identified merely by using other spurious noises, such as distinctive breathing patterns or external temporally-correlated noises, we also test on recordings from each human standing in the same positions as the RIRs, but while the speaker is silent. We randomly split the train and test data such that both have a balanced ratio of recordings from each human.

**Baselines** We use similar baselines to those used in Section 4.1, with the exception that we omit Linear Regression and our analytical Time of Arrival since they are both specifically suited to regression and localization, respectively. The VGGish-based models use the same structure as those in localization except that the last layer uses a Softmax to predict the likelihood scores for each class.

**Results** Table 5 shows the results. We see strong evidence that the actively-measured RIR indeed has important information for identification, and that the passively-recorded noises from the room or emitted by the humans while standing in presumed silence are generally not enough for our baselines to identify. Once again, our multichannel VGGish baseline consistently outperforms other baselines.

### 4.3 Human Detection

We investigate whether recordings from a room can be used to detect whether a single human is in the room or not. To do so, we compare recordings of a room with a human in it to those of the empty room. We observe that sine sweep-derived RIRs of an empty room are quite similar between trials. When a human is introduced into the room, the sine sweep-derived RIR becomes different enough to make the task of detecting the presence of a human from a sine sweep trivial. Instead, we focus on the task of using natural music signals to detect the presence of a human. We use data from our Treated Room, where we have 1,000 recordings, each of music played in the room with and without a human. We devise three different subtasks based on different inputs used to detect a human in the room: 1) an RIR derived from music, 2) a segment of the raw recording of the music, and 3) a segment of a recording of silence. Note that the first subtask requires having access to the time-synchronized signal that is being played in order to deconvolve out an estimate of the RIR, and the second does not assume such knowledge of the signal being played.

**Baselines**  We use the same baselines as in the identification task in Section 4.2, with the exception that the final output of the VGGish-based models is that of a Sigmoid rather than a Softmax.

**Results**  We show results for each of the different signals in Table 6. Once again, silence is not enough for our models to significantly outperform random guessing. Our multichannel VGGish baseline is able to detect a human in the room from these music-derived RIRs almost perfectly. However, knowledge of the signal being played seems to be essential to our baselines, since their performance on the music-derived RIR is much better than that on the raw music without knowledge of the underlying signal. In many real-world scenarios, access to the signal being played or emitted by the sound source is not available.

## 5   Limitations and Conclusion

We presented SOUNDCAM, a large dataset of unique room impulse responses (RIRs) and recorded music from a controlled acoustic lab and in-the-wild rooms with different humans in positions throughout each room. We have demonstrated that our dataset can be used for tasks related to localizing, identifying, and detecting humans. Our results show that while each of these tasks can be solved rather robustly when training on large datasets and testing on data from within the same distribution, there are still unsolved problems in generalizing to unseen humans as well as unseen rooms. These unsolved problems may hamper the practical real-world applications of these tasks.

The main limitations of our dataset are the breadth of environments, humans, and poses represented. While we collect data from two in-the-wild rooms, these rooms cannot possibly represent the wide distribution of rooms from different regions and socioeconomic backgrounds. The five humans in our dataset similarly cannot capture the diversity of age, sizes, and shapes of humans in the world. Finally, each human assumes the same pose facing the same direction to remove minor differences in pose as a variable, whereas in real-world applications, humans assume diverse poses as they go about their activities. While a wider diversity of human shapes and poses would be beneficial for widespread real-world applications, SOUNDCAM is a significant first step in developing such applications.

As future work, we first plan to augment the diversity of humans and their poses in our dataset. Our dataset currently includes more annotations than needed for our tasks, such as 3D scans of each room, and could thus be used for additional novel tasks, such as prediction of RIR from simulation. We also hope to augment the dataset to vary the number of people in the room, so that the dataset could be used for tasks of quantifying, identifying, or locating multiple people in a single room using sound. Finally, the failure of our baselines in learning from raw audio signals rather than RIRs suggests additional applications in estimating room acoustics from arbitrary source signals or even techniques which do not assume knowledge of the source signal, such as blind deconvolution.

**Acknowledgments.**   We thank Mert Pilanci and Julius O. Smith for valuable discussions. This work is supported in part by NSF CCRI #2120095, ONR MURI N00014-22-1-2740, Adobe, Amazon, and the Stanford Institute for Human-Centered AI (HAI). The work was done in part when S. Clarke was an intern at Adobe.

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

# Appendix

The supplementary materials consist of:

- A. Details of the rooms used in the dataset.
- B. Details of the baseline implementations and their hyperparameters.
- C. Additional results from experiments in the human localization tasks.
- D. Details on the procedures for data collection and preprocessing.
- E. A datasheet with important metadata about the dataset.

## A  Room Details

We include photos and further details of each room.

### A.1  Acoustically Treated Room

The Treated Room is rectangular, approximately 4.9×5.1 meters and 2.7 meters in height. Each of its walls are covered in 15-cm-thick melamine foam panels, each having a noise reduction coefficient (NRC) of 1.26. The ceiling tiles are fiberglass with an NRC of 1.0, and the floor is carpeted with standard office carpet. Photos of the room in its two different configurations, empty and with fabric divider panels, are shown in Figures 4and 5, respectively. Diagrams of the positions of the Human 1 subsets from the room in each configuration are shown in Figure 6. The microphones and speaker remained in the same positions for both subsets.

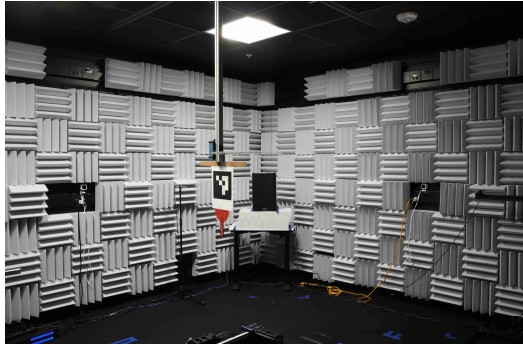 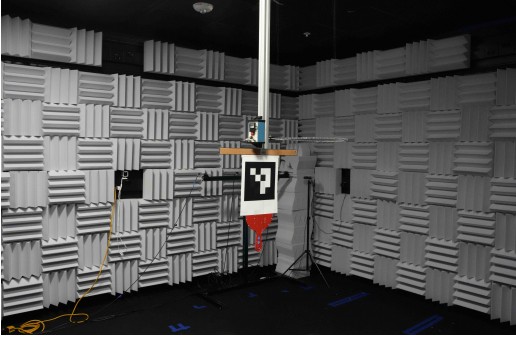

Figure 4: Images from the Treated Room in its empty configuration.

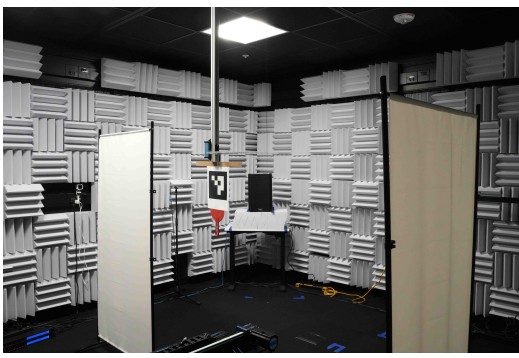 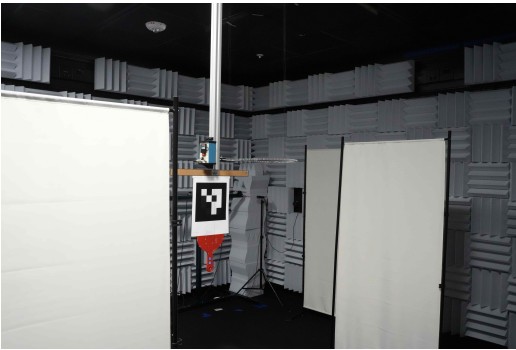

Figure 5: Images from the Treated Room in its configuration with fabric panels.

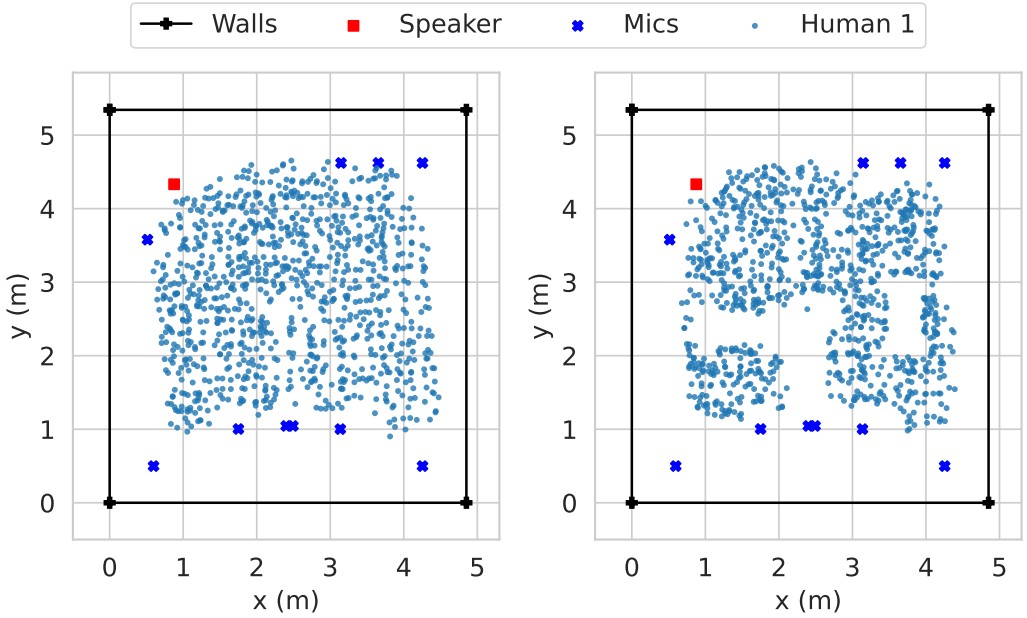

Figure 6: Diagrams of the positions of speaker, microphones, and Human 1 in Treated Room subsets in (**Left**) empty configuration (representing both sine sweep and music subsets) and (**Right**) configuration with fabric divider panels.

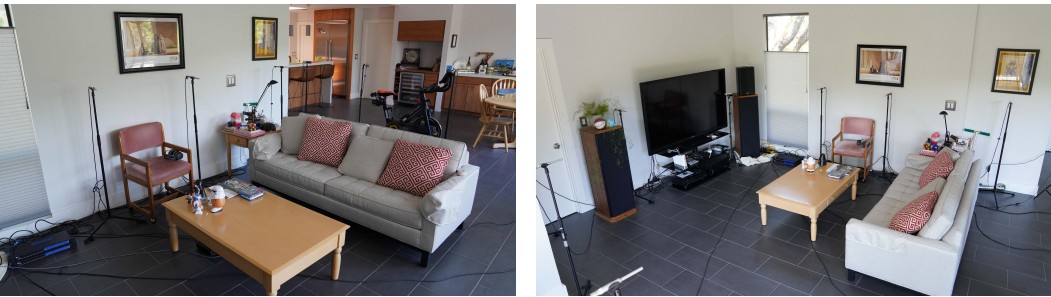

Figure 7: Images from the Living Room.

## A.2 Living Room

The Living Room is in a real household with an open layout, *i.e.*, the room does not have specific walls delineating it from parts of the rest of the house, including a kitchen and a stairway. The floor is covered in tile, and nearby walls and the vaulted ceiling are composed of drywall. The room contains a television screen and entertainment system, a coffee table, and a fabric sofa, with other furniture outside the boundaries of the area used for data collection also affecting the acoustic signature. In addition to the photo of the room and the diagram of the microphone, speaker, and human positions included in Figure 2 of the main manuscript, Figure 7 shows additional photos of the room.

## A.3 Conference Room

The Conference Room is rectangular, approximately 6.7×3.3 meters and 2.7 meters in height. Three of its walls are composed of drywall, while the remaining wall (on a short side of the rectangle) is mostly composed of glass, with drywall trim and a wooden door with a glass window. The ceiling is closed with standard office ceiling tiles, and the floor is covered with standard office carpet. The room has a large monitor screen on one wall, a large whiteboard on another wall, and a long wooden table pushed against the wall with the monitor, with wheeled chairs surrounding it. Photos from the Conference Room are shown in Figure 8, and the diagram of speaker, microphone, and human positions in the room are shown at the right of Figure 9.

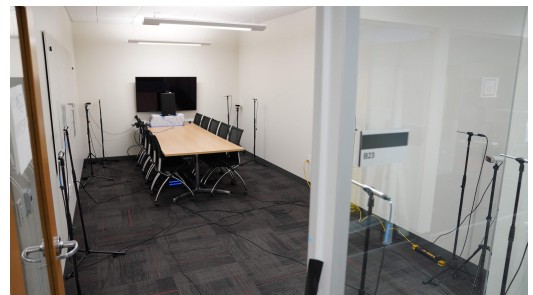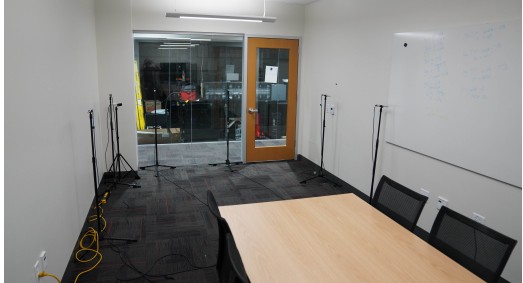

Figure 8: Images from the Conference Room.

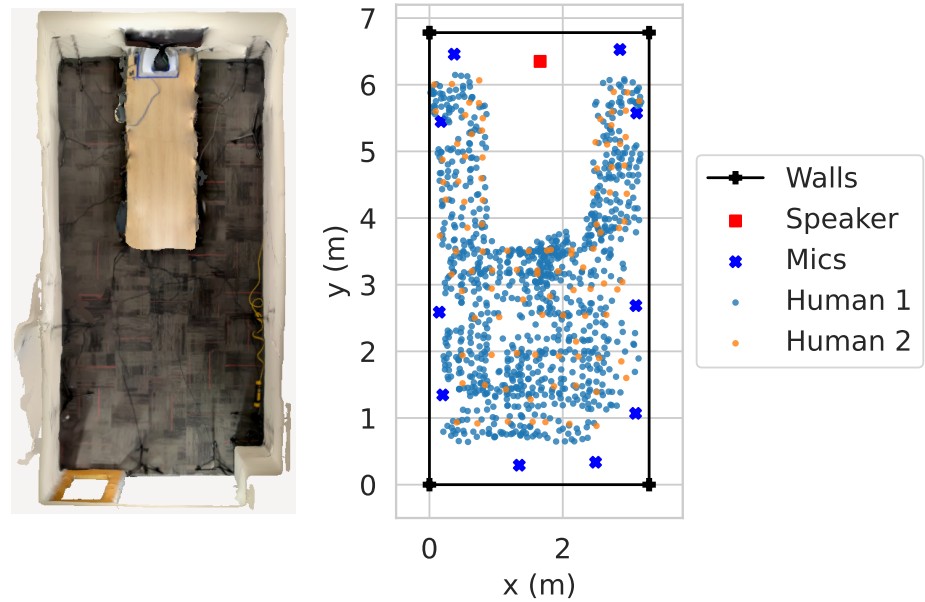

Figure 9: (**Left**) An aerial view of a 3D scan of the Conference Room. (**Right**) Diagram of the microphone, speaker, and unique positions of Human 1 in the Conference Room.

## A.4 RT60 Reverberation Times

The reverberation time for each room is measured from the empty room's impulse response, from which we calculate the time for the sound to decay by 60 dB (RT60) using the Schroeder method [Schroeder, 1965]. The RT60 measurements for each of the rooms/configurations are shown in Table 7. Since there are 10 microphones from which RT60s can be measured, we report the mean, minimum, and maximum RT60 across the 10 microphone locations.

|  | Mean RT60(s) | Min RT60(s) | Max RT60 (s) |
|---|---|---|---|
| Treated Room | 0.158 | 0.112 | 0.264 |
| Treated Room w/ Panels | 0.158 | 0.111 | 0.248 |
| Living Room | 1.121 | 1.022 | 1.170 |
| Conference Room | 0.581 | 0.541 | 0.608 |

Table 7: Overall RT60 Reverberation Times for each of the rooms/configurations. The second column shows the minimum RT60 across all microphones. The third column shows the maximum RT60 across all microphones.

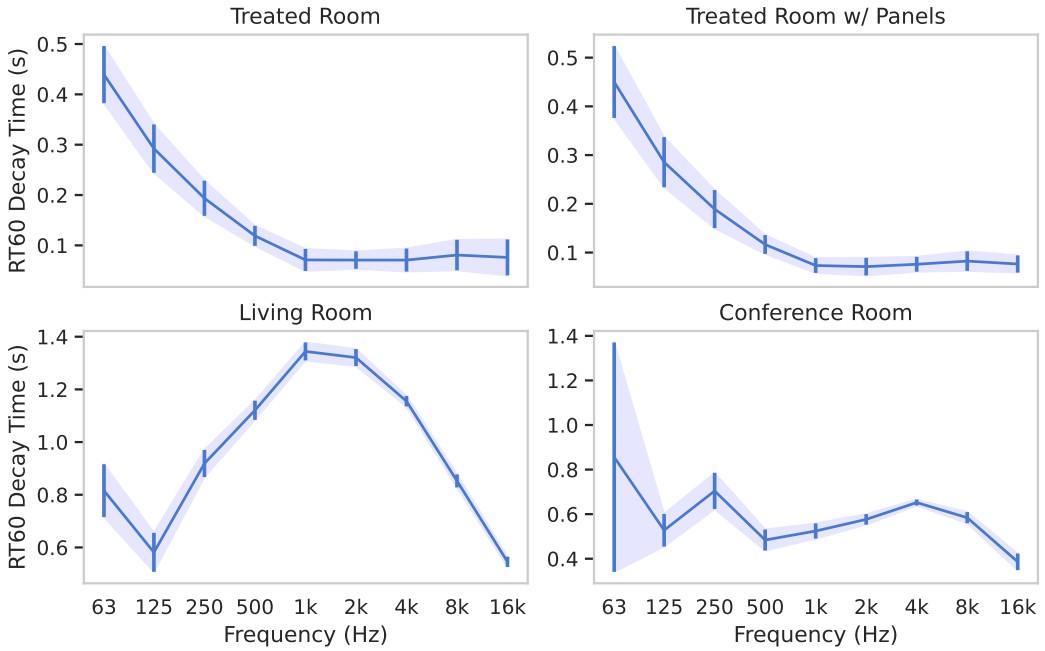

Figure 10: Octave-band RT60 reverberation time measurements from each of our four rooms/configurations. The solid line indicates the average RT60 for each frequency band across all microphone locations, while the shaded region represents the standard deviation.

| | Sweep Level (dB) | Silence Level (dB) | Content of Silence |
|---|---|---|---|
| Treated Room | -3.76 | -42.76 | Microphone noise |
| Living Room | -2.57 | -44.67 | Microphone noise, occasional vehicle |
| Conference Room | 0.00 | -30.60 | Microphone noise, ventilation noise |

Table 8: RMS levels of the recordings during the sweep (first 10 seconds) and silence (last 2 seconds). Levels are measured in dB, relative to the sweep level in the conference room, which was the loudest. Lower dB levels mean the silence is quieter relative to the sound being played.

In addition, we provide octave-band RT60s in Figure 10. We observe that reverberation times differ significantly across frequency bands, and that the relationship between frequency and reverberation time is different for each room.

### A.5 Silence Levels and Content

The last 2 seconds of every recording are presumed to be silent, since recording continues for 4 seconds after the sweep/music signal plays from the speaker, and the longest RT60 measured in any of our rooms at any frequency was under 1.5 seconds. Table 8 shows the RMS level when the sweep is playing (first 10 seconds), relative to the RMS level of the presumed silence (last 2 seconds). RMS levels are measured relative to the sweep level in the conference room. The levels are averaged across all microphone recordings.

## B   Baseline Details and Hyperparameters

We include further details of each baseline in addition to important hyperparameters.

### B.1   Levels-based Baselines

For all baselines based on volume levels, each room impulse response from each microphone was truncated to two seconds, and condensed into a single number representing the RMS volume level

of the room impulse response. Thus, for baselines that use all 10 microphones, each data point is summarized by 10 features, one for each microphone.

Each feature for the data points is normalized to have a mean of 0 and a standard deviation of 1 across the training set. We use the validation set of 100 datapoints to determine the best-k for k-nearest neighbors, with k ranging from 1-100, and report the average error in centimeters on the test set. The linear regression baseline does not use the validation set, and fits a linear model that minimizes squared error on the training set.

For baselines using raw music recordings (as opposed to room impulse responses, or room impulse responses estimated from music), a 2-second interval from the middle of the recording (from 5 to 7 seconds) is taken instead.

For baselines using silence, we examine the raw recording of the sine sweep that was used to measure the room impulse response. The sine sweep plays for 10 seconds in this recording, followed by 4 seconds of silence. For our baselines, the last two seconds of this raw audio recording are taken and assumed to be silent, since the longest RT60 Reverberation Time of any of our rooms at any frequency was measured to be 1.5 seconds.

### B.2 Deep Neural Network-based Baselines

Because the pretrained VGGish model from [Hershey et al., 2017] was previously trained on AudioSet [Gemmeke et al., 2017], the authors had designed the model around some key assumptions about the input audio, including the sample rate and the number of channels. We accordingly conducted some preprocessing of our input audio before before passing it to the pretrained VGGish. First, we downsample the audio from 48 kHz to 16 kHz, to match the sampling rate on which the VGGish was pretrained. Next, if the input includes audio from multiple microphones, we collapse the audio to a single channel by taking the mean across all channels. We then truncate the input audio to 30950 samples (or 1.93 seconds) to match the pretrained network's input size. The audio is then transformed into a log-mel spectrogram, which is passed to a deep convolutional neural network that generates a 128-dimensional representation of the audio. We train a 3-layer, fully connected neural network with ReLU activations following hidden layers on these embeddings, to predict an $x, y$ location, a vector of class probabilities, or a single number representing the probability of human presence in the room. The dimension of each hidden layer is 256.

For the multichannel VGGish baselines, we do not downsample or reduce the channels of the input audios as part of the preprocessing and instead compute the log-mel spectrogram on each input channel at full sample rate. For these baselines, the input to the neural network is an $N$-channel tensor of stacked spectrograms from each channel, where $N$ is the number of audio channels. The neural network for multichannel VGGish is based on the same overall structure as that of the pretrained VGGish in terms of the number of layers and size of the convolutional kernels at each layer, other than the first convolutional layer which is resized to accommodate inputs of more than one channel. We randomly initialize all weights at the beginning of training.

Thus, the key differences between the multichannel VGGish and the pretrained VGGish are that 1) the weights in the CNN of the pretrained VGGish's weights are frozen to those from AudioSet, while the weights of the multichannel VGGish are not, and randomly initialized, 2) the input is collapsed to a single channel in the pretrained VGGish via mean reduction, and 3) the input is reduced to a sampling rate of 16 kHz in the pretrained VGGish while the multichannel VGGish maintains the full sample rate (48kHz).

All VGGish-based models are trained for 1000 epochs using the Adam optimizer with a learning rate of $1 \times 10^{-4}$ and a batch size of 32.

### B.3 Time of arrival Baseline

Our time of arrival baseline is based on principles described in [Yang et al., 2022], which contains more details about the principles we used. As a brief summary, the room's impulse response is modeled as the sum of two room impulse responses - the impulse response of the empty room, and the impulse response resulting from the additional reflections induced by the person in the room.

To isolate the reflections generated by the person, we subtract the empty room impulse response from the impulse response obtained when the room has a person in it. This provides us with a time-domain

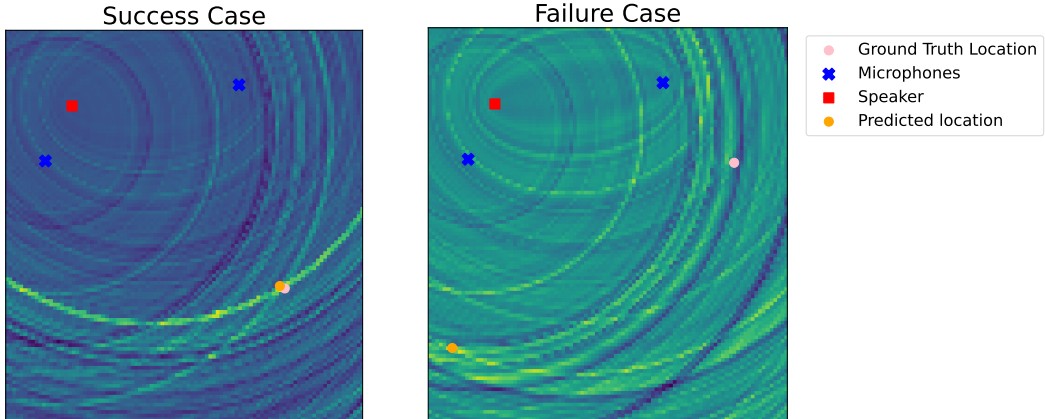

Figure 11: Visualizations of the intersecting spatial heatmaps from the Time-of-Arrival baseline, using the positions of the two microphones closest to the speaker from two-channel impulse responses in the Treated Room. (**Left**) A success case and (**Right**) a failure case.

|  | Time of arrival | Center Prediction |
|---|---|---|
| Treated Room | 96.2 | 136.0 |
| Treated Room w/ Panels | 89.3 | 144.3 |
| Living Room | 225.9 | 176.6 |
| Conference Room | 136.0 | 159.0 |

Table 9: Median error in centimeters for the time of arrival method and the center-of-room prediction method. The results are using 4-channel RIRs measured from sine sweeps recorded in each of the rooms/configurations.

representation of the person's reflections, which is referred to by Yang et al. [2022] as the "pose kernel". This representation can be mapped to a spatial encoding of where the person might be standing based on the speed of sound and the positions of the sound source and listener.

In order to obtain a measurement of the empty room impulse response, we compute the average of the 100 room impulse responses measured in the empty room. Averaging in the time domain is possible due to the time-alignment procedure described in Appendix D.1.

In order to determine a person's location from one- or many-channel room impulse responses, we compute a pose kernel for each impulse response. We take the envelope of each pose kernel, and use them to compute a spatial heatmap of possible locations. These spatial heatmaps are computed on an $xy$ grid with a resolution of 5 cm, covering the space within the walls of the room.

When the impulse responses are from multiple channels, these spatial heatmaps are intersected by multiplying them together. The predicted location is the location with the maximum value on the resultant heatmap.

A success and a failure case for the time of arrival method are visualized in Figure 11.

It is worth noting that failure cases for this baseline are often catastrophic, resulting in large errors which inflate the mean error. In Table 9 we provide the median test error of the time of arrival baseline for each room. The results are shown for the 4 microphone case, where RIRs are estimated from sine sweeps. The median errors are from the exact same experiments as those in Table 2. We compare it to the method that predicts the center of the room each time. As we can see, the time of arrival baseline outperforms center-of-room guessing by a substantial margin in the Treated Room and the Treated Room with Panels, and performs worse than it in the Living Room. It slightly outperforms center-of-room guessing in the Conference Room. This suggests that this method is based on delicate assumptions and may not be robust to the challenges of uncontrolled environments in the wild. Since this approach has the advantage of not requiring training examples, these results motivate future work to make this approach more robust to reverberant rooms.

| | Number of Microphones | | | |
|---|---|---|---|---|
| | 10 | 4 | 2 | 1 |
| kNN on Levels | 26.0 (35.9) | 64.0 (53.4) | 117.2 (80.3) | 150.6 (73.0) |
| Linear Regression on Levels | 97.7 (62.6) | 116.3 (63.1) | 129.6 (67.7) | 153.2 (71.6) |
| VGGish (pretrained) | 101.1 (76.5) | 102.9 (75.6) | 92.5 (69.4) | 120.0 (87.5) |
| VGGish (multichannel) | **24.0** (29.9) | **20.7** (28.4) | **27.5** (32.4) | **44.4** (42.4) |
| Time of Arrival | 181.9 (105.8) | 152.7 (87.4) | 218.1 (114.4) | 382.9 (134.7) |

Table 10: Localization error of each model using sine sweep-based RIRs from varying numbers of microphones in the Conference Room. Errors are in centimeters and "mean (stdev)" format.

| | Number of Microphones | | | |
|---|---|---|---|---|
| Train Base → Test w/ Panels | 10 | 4 | 2 | 1 |
| kNN on Levels | 146.3 (77.4) | 162.6 (109.2) | 149.1 (85.1) | **135.7** (55.2) |
| Linear Regression on Levels | **126.3** (46.0) | 131.8 (53.3) | 138.1 (48.3) | 138.3 (47.6) |
| VGGish (pretrained) | 163.1 (85.6) | 168.9 (74.4) | 157.3 (80.5) | 163.8 (75.9) |
| VGGish (multichannel) | 132.6 (107.8) | **126.0** (97.4) | **135.1** (71.2) | 163.6 (75.0) |
| Train w/ Panels → Test Base | | | | |
| kNN on Levels | 182.5 (111.0) | 170.5 (88.5) | 175.4 (86.7) | 135.7 (56.7) |
| Linear Regression on Levels | 367.8 (274.2) | 151.4 (92.3) | **134.9** (59.3) | **135.0** (49.3) |
| VGGish (pretrained) | 219.3 (96.8) | 146.2 (72.8) | 151.4 (73.0) | 169.2 (90.6) |
| VGGish (multichannel) | **114.8** (68.7) | **133.9** (47.8) | 175.1 (91.2) | 143.0 (77.2) |

Table 11: Localization error of each model when trained on recordings of a human from the treated room in one arrangement and tested on recordings from the same human in the same room in the other arrangement. Errors are in centimeters with "mean (stdev)" format.

## B.4 Microphone Ablations

For baselines experiments that only use one microphone, the furthest microphone from the speaker is selected. For baseline experiments that use two microphones, the closest microphone to the speaker and the furthest microphone from the speaker are selected. For baselines that use four microphones, microphones from each corner of the room are selected.

## B.5 Resources used

In order to run our baseline experiments, we used the GPU clusters provided to our research group. The kinds of GPUs are NVIDIA GeForce RTX 2080 Ti, Titan RTX, GeForce RTX 3090, A40, A5000, and A6000. Altogether, the baseline experiments shown in the paper consumed approximately 300 GPU hours.

## C   Additional Baseline Results

We include additional results from the localization task to first show the performance of each baseline on generalizing to a different configuration of the room and next show the effects of downsampling recordings on our multichannel VGGish baseline.

## C.1   Additional Results on Human Localization

We show additional quantitative results from experiments described in Section 4.1 of the main manuscript. In addition to the results shown in Table 2 of the main manuscript, the results of our baselines on localizing a human from sine sweep-based room impulse responses (RIRs) within the Conference Room are shown in Table 10. The results of training on sine sweep-based RIRs from the empty Treated Room and testing on sine sweep-based RIRs from the same human in the Treated Room with fabric divider panels, and vice versa, are shown in Table 11.

| Number of Data Points | Localization Error (cm) |
|---|---|
| 25 | 166.5 (61.2) |
| 50 | 112.9 (78.3) |
| 100 | 67.0 (51.1) |
| 200 | 52.2 (37.9) |
| 300 | 34.3 (27.0) |
| 400 | 42.1 (33.6) |
| 500 | 31.5 (24.1) |
| 800 | **27.9** (22.0) |

Table 12: Localization error of the Multichannel VGGish model in the living room when trained on various numbers of data points. The models are trained on 10-channel RIRs. Errors are in centimeters with "mean (stdev)" format.

| Model | Number of Microphones | | | |
|---|---|---|---|---|
| | 10 | 4 | 2 | 1 |
| VGGish (multichannel, 48 kHz) | **18.1** (13.1) | **17.2** (16.2) | **20.4** (15.4) | 71.6 (50.4) |
| VGGish (multichannel, 16 kHz) | 18.6 (15.4) | 21.8 (16.1) | 26.8 (23.1) | **49.4** (54.3) |

Table 13: Localization errors of the multichannel VGGish with the full 48 kHz sampling rate, and the multichannel VGGish when the input audio is downsampled to 16 kHz. The models are trained and evaluated on RIRs from the Treated Room. Errors are in centimeters with "mean (stdev)" format.

| RIRs from Treated Room, RIR | Number of Microphones | | | |
|---|---|---|---|---|
| | 10 | 4 | 2 | 1 |
| VGGish (multichannel, 48 kHz) | **82** | **81** | **76** | **64** |
| VGGish (multichannel, 16 kHz) | 63 | 60 | 51 | 42 |

Table 14: Classification Accuracy (%) in identifying among five humans using RIRs in the Treated Room for the multichannel VGGish, using the full 48 kHz sampling rate and downsampling to 16 kHz.

In addition, we perform a data ablation study testing sample efficiency on the localization task for the multi-channel VGGish in the living room. Training on just 100 10-channel RIRs, we are able to localize the human to within 68 cm in the test set. These results are shown in Table 12.

### C.2 Effect of Sample Rate on Baseline Performance

Since VGGish (pretrained) is pretrained on audio recordings with a sample rate of 16 kHz, we were interested in seeing if the higher, 48 kHz sample rate used in our dataset is justified by isolating the influence of changing sample rate on our best-performing model, the multichannel VGGish. Table 13 shows results for the human localization task. We see that the model trained and evaluated on 48 kHz RIRs outperforms the the model trained and evaluated on RIRs downsampled to 16 kHz in all conditions except for the single-microphone case. Table 14 shows the classification accuracy for both versions of the VGGish on the 5-person human identification task. These results both suggest that the higher sample rate is an important advantage for extracting task-relevant insights from these recordings.

## D    Additional Details on the Data Collection Procedure and Preprocessing

We describe aspects of our data collection procedure and processing in more details, including the logistics of how the human is directed to move between sine sweeps (App. D.1), the estimation of the room impulse responses (RIR) from recordings (App. D.2), and collecting recordings of music (App. D.3).

### D.1 Collection Procedure

Room impulse responses in the dataset are measured by playing and recording sine sweeps. The details of how room impulse responses are obtained from these recordings are described in (App. D.2). If there is a person in the room, they are directed (by a beep) to select a location to stand in before the sine sweep begins playing. Another beep tells them that they should stand still, and two seconds later, their location and pose are captured by the RGBD cameras in the room. Then, the sine sweep is played from the speaker and recorded from the 10 microphones simultaneously. If we are capturing recorded music as well, a short pause is given after the sine sweep is finished recording, and then a 10-second music clip is played and recorded as well. Below, we have details on the processing techniques used to obtain the room impulse response, and to ensure our data is properly time-aligned.

### D.2 Measuring the Room Impulse Response

In order to measure a room impulse response, we had the speaker play a logarithmic sine sweep from 20 Hz to 24,000 Hz for 10 seconds, followed by 4 seconds of silence. This sine sweep was recorded from each of the ten microphones. At the same time that the sine sweep is being sent from the audio interface to the speaker, a loopback signal is also being sent from the audio interface's output to one of its inputs. This loopback signal is used to estimate and correct for the latency in the system.

To compute the room impulse response $r[t]$, we take

$$r[t] = IFFT\left(\frac{FFT(a[t])}{FFT(l[t])}\right)$$

Where $FFT$ and $IFFT$ are the Fast-Fourier Transform and its inverse, $a[t]$ is the digital recording of the sine sweep, and $l[t]$ is the digital loopback signal. Observe that we deconvolve the loopback signal from the recording, instead of deconvolving the source signal sent to the speaker from the recording. The loopback signal is assumed to be the same as the source signal, but delayed in time in an amount equal to the latency of the system. Deconvolving from a delayed copy of the source signal instead of directly from the source signal corrects for the delay in the system. The last 0.1 seconds of the 14-second room impulse response is removed to eliminate anti-causal artifacts.

In case cases where the room impulse responses needs to be estimated from music, the procedure is identical, but the sine sweep is replaced by music.

### D.3 Recording Music

We selected 1000 different clips of music from the 8 top-level genres included in the small version of the Free Music Archive dataset provided by Defferrard et al. [2016]. The genres included are "International", "Pop", "Rock", "Electronic", "Hip-Hop", "Experimental", "Folk", and "Instrumental". The clips are selected randomly, and the same number of clips are selected from each genre. Each clip is truncated to 10 seconds, and padded by four seconds of silence at the end to allow for the music to decay. In order to keep the volume level between clips consistent, the music clip being played is normalized to have the same root mean square across all data instances. The music is played by the speaker and recorded at the same time from each of the 10 microphones. In order to correct for the latency in the system, a loopback signal is recorded at the same time. The loopback signal is thresholded to determine when the music began playing from the interface, which is used to estimate the delay in the system. All music recordings are adjusted by this delay and saved in a separate file. The raw music recordings (without adjustment) are included, as well as the loopback signals.

## E  SOUNDCAM Datasheet

Following the guidelines suggested in Gebru et al. [2021], we document details of the SOUNDCAM dataset below.

### E.1 Motivation

**For what purpose was the dataset created?** Was there a specific task in mind? Was there a specific gap that needed to be filled? Please provide a description.

We created SOUNDCAM to develop methods for tracking, identifying, and detecting humans using room acoustics. SOUNDCAM allows for the training and testing of these methods in controlled and real-world environments. We describe these three tasks and provide baseline methods for them. Our results demonstrate that all of these tasks are unsolved under certain conditions. To our knowledge, no other publicly available dataset can be used to test methods in tracking, identifying, or detecting humans using acoustic information from real rooms.

**Who created the dataset (e.g., which team, research group) and on behalf of which entity? (e.g., company, institution, organization)?**

SOUNDCAM was created by a collaboration among the authors on behalf of their organizations, Stanford University and Adobe Inc.

**Who funded the creation of the dataset?**

The National Science Foundation and Adobe Inc.

**Any other comments?**

None.

## E.2 Composition

**What do the instances that comprise the dataset represent? (e.g., documents, photos, people, countries)?** Are there multiple types of instances (e.g., movies, users, and ratings; people and interactions between them; nodes and edges)? Please provide a description.

The types of data instances in SOUNDCAM can be split into four categories:

1. **Empty Room Impulse Responses**. These instances contain a measurement of the room impulse response of an empty room (*i.e.* without a human in it), as measured from 10 microphones.

2. **Room Impulse Responses, From Rooms With a Human in Them**. These instances contain a measurement of the room impulse response in a room, with a person standing at a specific, measured location. Each instance contains 1) a measurement of the room impulse response as measured from 10 microphones; 2) a measurement of the person's location in the room, determined by the pelvis joint location as estimated by three RGBD cameras; 3) the locations of the 17 joints given by the Azure Kinect's body tracking tool [Microsoft, 2022]; and 4) the anonymized identifier of the person present in the room.

3. **Recordings of Music in an Empty Room**. These instances contain a recording of a 10-second music clip playing in an empty room, as measured from 10 microphones. We vary the music clips between datapoints, selecting them from [Defferrard et al., 2016]. The source music clip is included in each datapoint, as well as a number identifying the source music clip.

4. **Recordings of Music, in a Room With a Human Standing in It**. These instances contain a recording of a 10-second music clip playing in a room with a person is standing in it. Each instance contains 1) a recording of the music playing in the room as measured from 10 microphones; 2) a measurement of the person's pelvis location; 3) the locations of the 17 joints given by the Azure Kinect's body tracking tool; 4) the source music clip, as well as a number identifying it; and 5) the anonymized identifier of the person present in the room. The same 1000 music clips that are used across rooms, as well as when we record music in the empty room.

In addition, we provide a 3D scan of the each room in which we collected data, along with the 3D locations of each microphone and of the speaker. We also include calibrations for each of the 10 microphones we used for measurements. We also include a rough, anonymity-preserving scan of each person represented in the dataset in the same poses they assumed while we collected the recordings.

**How many instances are there in total (of each type, if appropriate)?**

We summarize this in Table 1 of the main manuscript.

**Does the dataset contain all possible instances, or is it a sample (not necessarily random) of instances from a larger set?** If the dataset is a sample, then what is the larger set? Is the

sample representative of the larger set (e.g. geographic coverage) If so, please describe how this representativeness was validated/verified. If it is not representative of the larger set, please describe why not (e.g., to cover a more diverse range of instances, because instances were withheld or unavailable).

The dataset represents some imperfect sampling at a number of levels. First, the locations of the humans in the room are sampled from the set of all possible feasible locations that someone could be standing in the room, though each human may have been more biased toward certain regions. We validate how representative each sample is by providing visualizations of the rooms and the corresponding human positions in Figures 2, 6, and 9. The five humans who volunteered as subjects could not possibly represent the diversity of the global population from which we sampled them. We would need at least an order of magnitude more distinct humans to approach this ideal. We sampled the music clips for our recordings from the 'fma-medium' version of the Free Music Archive Defferrard et al. [2016] dataset, sampling uniformly within each of the eight major genres.

**What data does each instance consist of?** "Raw" data (e.g., unprocessed text or images) or features? In either case, please provide a description.

We provide a description of each data instance's annotation in an answer to a previous question. Both "raw" data and preprocessed data are included in the dataset. For instance, each room impulse response is measured by playing a sine sweep from the speaker. The recording of this sine sweep is also included in the dataset alongside the RIR produced through preprocessing this recording. More detailed descriptions of the "raw" data included in the dataset are discussed below in Appendix E.4.

**Is there a label or target associated with each instance?** If so, please provide a description.

For the localization task, the labels are the planar $x, y$ location of the human's pelvis joint. For the detection task, the label is if the person is in the room or not. For the identification task, the label is the person's identity. We document all of these labels for each data instance.

**Is any information missing from individual instances?** If so, please provide a description, explaining why this information is missing (e.g., because it was unavailable). This does not include intentionally removed information, but might include, e.g., redacted text.

Everything is included for each individual instance. No annotation is missing for any particular instance.

**Are the relationships between individual instances made explicit (e.g., users' movie ratings, social network links)?** If so, please describe how these relationships are made explicit.

Yes, across each dimension of potential relationships, the relationships are made explicit by the annotations. For example, each of the 1000 song clips used is given a number, so data points using the same song clip can be matched. Clips with the same human in the room can similarly be matched.

**Are there recommended datasplits (e.g., training, development/validation, testing)?** If so, please provide a description of these splits, explaining the rationale behind them.

The train/valid/test split used for our baseline experiments are provided as a part of the dataset. These splits were done randomly, in a 80/10/10 fashion. For generalization experiments, we recommend holding out all of the data from the human, room, or signal to which the generalization is being tested.

**Are there any errors, sources of noise, or redundancies in the dataset?** If so, please provide a description.

See Appendix E.4 below.

**Is the dataset self-contained, or does it link to or otherwise rely on external resources (e.g., websites, tweets, other datasets)?** If it links to or relies on external resources, a) are there guarantees that they will exist, and remain constant, overtime; b) are there official archival versions of the complete dataset (i.e., including the external resources as they existed at the time the dataset was created); c) are there any restrictions (e.g., licenses,fees) associated with any of the external resources that might apply to a dataset consumer? Please provide descriptions of all external resources and any restrictions associated with them, as well as links or other access points, as appropriate.

The dataset is self-contained. For each of the song clips used in the dataset, we provide the song ID from the Free Music Archive as well as the song's genre.

**Does the dataset contain data that might be considered confidential (e.g., data that is protected by legal privilege or by doctor-patient confidentiality, data that includes the content of individuals' non-public communications)?** If so, please provide a description.

The dataset does not contain confidential information.

**Does the dataset contain data that, if viewed directly, might be offensive, insulting, threatening, or might otherwise cause anxiety?** If so, please describe why.

The dataset does not include any offensive or insulting aspects to our knowledge. However, it is possible that a small number of the songs we randomly sampled from the Free Music Archive Defferrard et al. [2016] may contain mildly crude language of which we are not aware. None of our subjects notified us as such.

**Does the dataset identify any subpopulations (e.g., by age, gender)?** If so, please describe how these subpopulations are identified and provide a description of their respective distributions within the dataset.

The dataset does not identify any subpopulations. The humans used in the dataset are between the ages of 22 and 30, between 157 cm and 183 cm tall, and represent a diverse set of genders and ethnicities. However, we do not specifically identify the subpopulations to which any individual subject belongs.

**Is it possible to identify individuals (i.e., one or more natural persons), either directly or indirectly (i.e., in combination with other data) from the dataset?**

No, the 3D scans of each individual are not detailed enough to identify any of the people, and we omit RGB data which could be used to distinguish other features.

**Does the dataset contain data that might be considered sensitive in any way (e.g., data that reveals racial or ethnic origins, sexual orientations, religious beliefs, political opinions or union memberships, or locations; financial or health data; biometric or genetic data; forms of government identification, such as social security numbers; criminal history)?** If so, please provide a description.

The dataset does not contain such sensitive information.

**Any other comments?**

None.

## E.3 Collection Process

**How was the data associated with each instance acquired?** Was the data directly observable (e.g., raw text, movie ratings), reported by subjects (e.g., survey responses), or indirectly inferred/derived from other data (e.g., part-of-speech tags, model-based guesses for age or language)? If data was reported by subjects or indirectly inferred/derived from other data, was the data validated/verified? If so, please describe how.

The details of the data collection procedure are described in Appendix D.1. The data was directly observable.

**What mechanisms or procedures were used to collect the data (e.g., hardware apparatus or sensor, manual human curation, software program, software API)?** How were these mechanisms or procedures validated?

In order to collect our data, we used 10 omnidirectional Dayton Audio EMM6 microphones placed on microphone stands throughout the periphery of the room. The location of each microphone stand is shown in Figures 2, 6, and 9. These 10 microphones are routed to a synchronized pair of MOTU 8M audio interfaces, which simultaneously record and play the sine sweep, at a sampling rate of 48 kHz. Python's sounddevice module was used to play and record the audio.

**If the dataset is a sample from a larger set, what was the sampling strategy (e.g., deterministic, probabilistic with specific sampling probabilities)?**

The music was sampled uniformly from the 'fma-medium' version of the Free Music Archive Defferrard et al. [2016].

Each human's location within the room is sampled from the set of feasible standing locations inside of that room given all obstructions (for instance, furniture), and within the boundaries of the microphones and cameras. We instructed each subject to move in a serpentine fashion in even steps to cover the entire room as uniformly as possible. Throughout the collection process, we reviewed the subjects' coverage of the room frequently (every ∼50 datapoints) to ensure uniform coverage. The subjects made several passes across the room in each dataset, and we sometimes directed the subjects to move randomly. These procedures ensure uniform coverage while also reducing correlation between location and time.

**Who was involved in the data collection process (e.g., students, crowdworkers, contractors) and how were they compensated (e.g., how much were crowdworkers paid)?**

All of those involved in the data collection process were students at Stanford University, who graciously volunteered their time to us.

**Over what timeframe was the data collected?** Does this timeframe match the creation timeframe of the data associated with the instances (e.g., recent crawl of old news articles)? If not, please describe the timeframe in which the data associated with the instances was created. Finally, list when the dataset was first published.

We collected the dataset between May 5, 2023, and June 4, 2023, which should be commensurate with the timestamps on the raw files. Preprocessed files and compressed versions of files may have later timestamps.

**Were any ethical review processes conducted (e.g., by an institutional review board)?**

The exemption specified in CFR 46.104 (d) (3) (i) (A) applies to our research. The exemption states that:

(i) Research involving benign behavioral interventions in conjunction with the collection of information from an adult subject through verbal or written responses (including data entry) or audiovisual recording if the subject prospectively agrees to the intervention and information collection and at least one of the following criteria is met:

(A) The information obtained is recorded by the investigator in such a manner that the identity of the human subjects cannot readily be ascertained, directly or through identifiers linked to the subjects;

**Did you collect the data from the individuals in question directly, or obtain it via third parties or other sources (e.g., websites)?**

We collected the data directly.

**Were the individuals in question notified about the data collection?** If so, please describe (or show with screenshots or other information) how notice was provided, and provide a link or other access point to, or otherwise reproduce, the exact language of the notification itself.

We notified the individuals that we were collecting data from them as we requested their participation. An example of the wording for this request is below.

**Did the individuals in question consent to the collection and use of their data?** If so, please describe (or show with screenshots or other information) how consent was requested and provided, and provide a link or other access point to, or otherwise reproduce, the exact language to which the individuals consented.

The individuals consented to the collection and use of their data. We asked all subjects if they would like to help us collect data, described the intentions of the project, and described the time and physical tasks we would require of them to participate. An example request is, "Would you have some time to help us collect data for a dataset on how room acoustics vary with humans in different positions? We would need you to stand silently in different positions in a room for about two hours total."

In addition, we have written documentation from all participants explicitly confirming their consent to the data collection procedure and to the data release. The participants acknowledged that they had the option to discontinue the data collection process at any time, and are also given the chance to opt out of the dataset and have their data removed.

The form that they signed can be viewed at `https://masonlwang.com/soundcam/ SoundCamConsentForm.pdf`

**If consent was obtained, were the consenting individuals provided with a mechanism to revoke their consent in the future or for certain uses?** If so, please provide a description, as well as a link or other access point to the mechanism (if appropriate)

We gave all subjects the choice to leave the data collection process and withdraw the data they provided at any time.

**Has an analysis of the potential impact of the dataset and its use on data subjects (e.g., a data protection impact analysis)been conducted?** If so, please provide a description of this analysis, including the outcomes, as well as a link or other access point to any supporting documentation.

We have not conducted a formal analysis, though we made design choices which were careful to consider subject privacy (*e.g.* omitting RGB images) and physical protection (*e.g.* providing hearing protection and allowing breaks as needed during data collection).

**Any other comments?**

None.

### E.4 Preprocessing/Cleaning/Labeling

**Was any preprocessing/cleaning/labeling of the data done (e.g.,discretization or bucketing, tokenization, part-of-speech tagging, SIFT feature extraction, removal of instances, processing of missing values)?** If so, please provide a description. If not, you may skip the remainder of the questions in this section.

We describe how the room impulse responses (RIRs) were derived in Appendix D.2 and describe how the music recordings were processed in Appendix D.3.

**Was the "raw" data saved in addition to the preprocessed/cleaned/labeled data (e.g., to support unanticipated future uses)?**

Yes, the raw data, including the raw recordings of the sine sweeps, the un-adjusted recordings of the music, and the recorded loopback signals, are provided alongside the main preprocessed dataset at `https://masonlwang.com/soundcam/`.

**Is the software used to preprocess/clean/label the instances available?**

Yes, the online repository for the source code is reachable by a link from the project page `https://masonlwang.com/soundcam/`

**Any other comments?**

None.

### E.5 Uses

**Has the dataset been used for any tasks already?** If so, please provide a description.

We have only used the dataset for the tasks described in this paper.

**Is there a repository that links to any or all papers or systems that use the dataset?**

Once others begin to use this dataset and cite it, we will maintain a list of selected uses at `https://masonlwang.com/soundcam/`

**What (other) tasks could the dataset be used for?**

Although we have documented the most apparent use cases, the dataset could be used for other tasks. For instance, it should be noted that the data collection process was conducted such that each data point collected with recorded music is paired with a room impulse response, where the human is standing at the same location. This could potentially be used to estimate room impulse responses from natural signals, like music. Furthermore, as mentioned in the Limitations and Conclusions, we collect scans of each room such that the dataset could be used to validate frameworks which simulate room acoustics from their 3D geometry.

**Is there anything about the composition of the dataset or the way it was collected and prepro-cessed/cleaned/labeled that might impact future uses?** For example, is there anything that a future

user might need to know to avoid uses that could result in unfair treatment of individuals or groups (e.g., stereotyping, quality of service issues) or other undesirable harms (e.g., financial harms, legal risks) If so, please provide a description. Is there anything a future user could do to mitigate these undesirable harms?

Users of the dataset should be aware of the distribution of shapes and sizes of the humans used to collect the data. Scans of each human are provided in the dataset itself, which should provide information about the sizes and shapes of the individuals in the dataset. As a brief summary, the humans used in the dataset are between the ages of 22 and 30, between 157 cm and 183 cm tall, and represent a diverse set of genders and ethnicities. However, there are only five humans, inescapably limiting the diversity from representing the human population. The dataset should be considered a starting point for developing methods to track, identify, and detect humans, but methods developed using SOUNDCAM should not be assumed to generalize perfectly to the wide variety of human shapes in the world.

**Are there tasks for which the dataset should not be used?** If so, please provide a description.

Yes, as mentioned in our Introduction, our dataset should not be used for tasks which involve covert tracking of non-consenting individuals for any applications in surveillance, etc. We believe that releasing this dataset to the academic community will inform more people to be cognizant of this potential misuse. Furthermore, we believe that we and the rest of the academic community can use our data to develop some robust defenses against such misuses.

**Any other comments?**

None

## E.6 Distribution

**Will the dataset be distributed to third parties outside of the entity (e.g., company, institution, organization) on behalf of which the dataset was created?** If so, please provide a description.

Yes, the dataset is publicly available and on the internet.

**How will the dataset will be distributed (e.g., tarball on website, API, GitHub)?** Does the dataset have a digital object identifier (DOI)?

The dataset can be reached from the project page `https://masonlwang.com/soundcam/`. It also has a DOI and can alternatively be reached at `https://doi.org/10.25740/xq364hd5023`.

**When will the dataset be distributed?**

The dataset is available for download at `https://masonlwang.com/soundcam/`. We will await reviewer feedback before announcing this more publicly.

**Will the dataset be distributed under a copyright or other intellectual property (IP) license, and/or under applicable terms of use (ToU)?** If so, please describe this license and/or ToU, and provide a link or other access point to, or otherwise reproduce, any relevant licensing terms or ToU, as well as any fees associated with these restrictions.

We distribute it under the MIT License, as described at `https://opensource.org/license/mit/`.

**Have any third parties imposed IP-based or other restrictions on the data associated with the instances?** If so, please describe these restrictions, and provide a link or other access point to, or otherwise reproduce, any relevant licensing terms, as well as any fees associated with these restrictions.

We collect all data ourselves and therefore have no third parties with IP-based restrictions on the data.

**Do any export controls or other regulatory restrictions apply to the dataset or to individual instances?** If so, please describe these restrictions, and provide a link or other access point to, or otherwise reproduce, any supporting documentation.

No.

**Any other comments?**

None

### E.7    Maintenance

**Who is supporting/hosting/maintaining the dataset?**

The dataset is hosted/maintained indefinitely in the Stanford Digital Repository at `https://doi.org/10.25740/xq364hd5023`, where both Mason Wang and Samuel Clarke have listed their contact information for inquiries.

**How can the owner/curator/manager of the dataset be contacted (e.g., email address)?**

Mason Wang can be contacted at ycda@stanford.edu, and Samuel Clarke can be contacted at spclarke@stanford.edu.

**Is there an erratum?**

There is currently no erratum, but if there are any errata in the future, we will publish them on the website at `https://masonlwang.com/soundcam/`

**Will the dataset be updated (e.g., to correct labeling errors, add new instances, delete instances)?** If so, please describe how often, by whom, and how updates will be communicated to users (e.g., mailing list, GitHub)?

To the extent that we notice errors, they will be fixed and the dataset will be updated.

**If the dataset relates to people, are there applicable limits on the retention of the data associated with the instances (e.g., were individuals in question told that their data would be retained for a fixed period of time and then deleted)?** If so, please describe these limits and explain how they will be enforced.

There are no limits on the retention of the data associated with the instances.

**Will older versions of the dataset continue to be supported/hosted/maintained?** If so, please describe how. If not, please describe how its obsolescence will be communicated to users.

We will maintain older versions of the dataset for consistency with the figures in this paper. These will be posted on a separate section on our website, in the event that older versions become necessary.

**If others want to extend/augment/build on/contribute to the dataset, is there a mechanism for them to do so?** If so, please provide a description. Will these contributions be validated/verified? If so, please describe how. If not, why not? Is there a process for communicating/distributing these contributions to other users? If so, please provide a description.

We will accept extensions to the dataset as long as they follow the procedures we outline in this paper. We will check recordings to make sure that recordings and room impulse responses are properly time-adjusted and have a good signal to noise ratio. We will label these extensions as such from the rest of the dataset and credit those who collected the data. The authors of SOUNDCAM should be contacted about incorporating extensions.

**Any other comments?**

The dataset will be maintained indefinitely on the Stanford Digital Repository.

