# OpenReview forum: "SoundCam: A Dataset for Finding Humans Using Room Acoustics"
_NeurIPS.cc/2023/Track/Datasets_and_Benchmarks — NeurIPS 2023 Datasets and Benchmarks Poster_

### Official Review · Reviewer_DvPP · 2023-07-04

**Rating:** 6
**Confidence:** 3

**Strengths:**

Active acoustic human localization and detection has not been studied much and the paper presents a new dataset for this purpose, which can be very useful and influential.

**Additional Feedback:**

-

**Clarity:**

In general the paper is quite clearly written, but the active/passive issue is problematic and creates confusion.

**Correctness:**

The paper uses acronym TDOA to denote the baseline method "time delay of arrival". This is very confusing since in localization literature, TDOA usually denotes "time difference of arrival", which is a very different method.

**Documentation:**

The following important details could have been useful to add to the paper:
-The silence recordings are described to consists of environmental sounds and human noises. Information about what those include and what are their levels would have been good, to understand to what degree the methods are actually doing passive/active localization.
-The RIRs are obtained by deconvolution. This is a standard procedure, but including information about what kind of deconvolution method was used would have been great, since the paper fundamentally relies on the estimated RIRs.

**Ethics:**

There are no ethical issue.

**Limitations:**

The issues related to the results have not been discussed.

**Opportunities For Improvement:**

The paper does not make it clear that this study and data are mostly about active localization. The discussion is switching between passive methods (for example the example in the introduction about cat meowing) and active methods. The datasets seems to mostly target active localization and detection, but there is one part of it (silence recordings) which is more about passive methods, but without explicitly stating that. The above issues make the target of the paper difficult to understand and will cause confusion since up to date, human localization, recognition, and detection has been mostly done with passive methods (e.g., speaker localization). A knowledgeable reader will eventually understand the difference between active and passive parts, but the manuscript could have made a much better job in stating these explicitly.

There is some shortcomings in the evaluation. First, it is not fully clear what is the localization error metric that is used. The tables explain the numbers to be the mean and std, but is the mean of the absolute error some someting else? If the mean is the absolute error, most of the baseline results are quite weak. For example, Linear Regression on Levels seems to lead to mean error which is similar to just predicting the center of each room. The TDOA method seems to give much worse results than that, indicating that there is severe problems in the TDOA method. I understand that the task is difficult so the results do not need to be highly accurate, but currently there so big gap between the VGGish (multichannel) and any other methods, so the validity of the results could have been explored better. Finally, many of the human detection results are below 50% (which is the random guess rate), indicating that there is some problems in the model training and selection.

**Relation To Prior Work:**

The paper discusses quite well previous studies. It should have cited some passive source localization studies to make it clear that in previous studies, human localization has been mostly done using passive methods.

**Summary And Contributions:**

The paper presents a dataset consisting of room impulse response measurements done with sine sweeps and music signals, including also the raw reverberant music signal, from three rooms which involve various positions of humans in the room, various locations of microphones. The paper presents baseline experiments related to human localization, recognition, and detection.

---

> ### Author Response · Authors · 2023-08-23
>
> Thanks for your valuable and detailed feedback.
>
> **Active vs Passive Localization**
>
> We updated the Introduction, Related Work, and Section 4.2 to make this distinction clearer.
>
> **Clarification on Error Metric**
>
> The error metric is the average Euclidean distance in cm between the ground truth location and the predicted location. We clarify this in Section 4.1.
>
> The multi-channel VGGish baseline indeed performs better than all others. We chose this baseline to see how deep-learning approaches could perform on these tasks, and to show potential shortcomings of current pretrained networks. The fact that multi-channel VGGish performs the best shows that deep-learning-based approaches might be well suited for these tasks. Although multi-channel VGGish does the best, other approaches have also shown promising results. For instance, kNN on levels using 10 microphones in the Treated Room achieved 30 cm of error.
>
> The poor performance of the linear regression on levels baseline shows that the problem is non-linear. We edited Section 4.1 to clarify this.
>
> **Content of Silence Recordings**
>
> In each room, we measure RMS levels for the sweep (first 10 seconds of the recording) and the presumed silence (last 2 seconds). These are averaged across all microphone locations, and measured relative to the sweep level in the Conference Eoom. More discussion of these results are in Appendix B.5.
>
> **Content of Silence Recordings**
>
> | | Sweep Level (dB) | Silence Level (dB) | Content of Silence |
> |-|-|-|-|
> | Treated Room | -3.76 | -42.76  | Mic noise |
> | Living Room | -2.57  | -44.67 | Mic noise, occasional vehicle |
> | Conference Room | 0.0  | -30.60 | Mic noise, ventilation noise  |
>
> **Deconvolution Method**
>
> We use FFT deconvolution to generate the room impulse responses. We include further details and a formula in Appendix E.2.
>
> **Time ~~Delay~~ of Arrival Baseline**
>
> Thank you for pointing this out the misnomer; we now use "time of arrival (TOA)" in our paper. This baseline was motivated by compelling prior work [Yang et al. 2022] and represents a traditional approach to the localization task, to see how well an analytical baseline that does not require training examples could do. We hope it serves as a benchmark for future analytical methods.
>
> Regarding concerns about this baseline's performance, below we show results we have added to Appendix C.3, which indicate that the baseline is somewhat effective as is and serves as a promising starting point for future work.
>
> Failure cases for this baseline are often catastrophic, resulting in large errors and inflating the mean error. Below, we provide the median test error of the TOA baseline, using 4 microphones, for each room (added to Appendix C.3). We compare it to always predicting the center of the room:
>
> | | TOA Median Error (4 mics) (cm) | Center-of-Room Median Error (cm)|
> |-|-|-|
> | Treated Room | 96.2| 136.0 |
> | Treated Room w/ Panels | 89.3 | 144.3  |
> | Living Room | 225.9 | 176.6 |
> | Conference Room | 136.0 | 159.0 |
>
> The TOA baseline substantially outperforms center-of-room guessing in the Treated Room with and without Panels, performs better than guessing in the Conference Room, and performs worse than guessing in the Living Room. These results show that the method is somewhat effective under controlled conditions, but its worse performance in the untreated rooms suggests that it is too sensitive to perform well in reverberant environments. We believe these results and the inherent advantages of an analytical method motivates this baseline as a starting point for future improvement.
>
> Appendix C.3 discusses the background and motivation behind the baseline, and source code is provided at https://github.com/maswang32/soundcam.
>
> **VGGish Detection Baseline**
>
> Thanks for raising concerns about our results in the detection tasks. We made a small adjustment to the experimental design of the binary detection baseline. The training and test sets were initially balanced completely randomly, such that methods could overfit to artifacts from specific songs in the training set. We modified the balance such that the training and test sets use a disjoint set of songs, with each set containing a positive and negative example for each represented song.
>
> All other experimental conditions remain the same. Updated results are in the table below and in Table 6 of the main manuscript.
>
> **Music RIR**
>
> | | 10 mics | 4 mics| 2 mics   | 1 mic  |
> |-|-|-|-|--
> | kNN on Levels| 57 | 58  | 56  | 54 |
> | VGGish (pretrained)| 64 | 62  | 70  | 80 |
> | VGGish (multichannel) | 99 | 100 | 100 | 98 |
>
> *Classification accuracy % in detecting the presence of a human in the Treated Room using a music-derived RIR*
>
> **Music Raw**
>
> || 10 mics | 4 mics| 2 mics  | 1 mic |
> |-|-|-|-|-|
> | kNN on Levels | 51 | 55 | 49 | 51 |
> | VGGish (pretrained) | 53 | 53 | 51 | 55 |
> | VGGish (multichannel) | 75 | 77 | 71 | 55 |
>
> *Classification accuracy % in detecting the presence of a human in the Treated Room using raw music*

---

> > ### Comment · Reviewer_DvPP · 2023-08-29
> >
> > Thank you for the explanations, these improve clearly the quality of the work. Some of the raised issues could have been explored further, but I am happy to update my recommendation.

---

> > > ### Author Response · Authors · 2023-08-30
> > > **Thank you**
> > >
> > > Thank you again for your helpful comments, which have helped us to improve our work.

---

> ### Author Response · Authors · 2023-08-28
>
> Thank you again for your valuable feedback. We hope our edits addressed your comments. Please let us know if there are any additional clarifications or edits you would advise. We would appreciate any additional feedback to further improve our draft as much as we can before the end of the rebuttal period on the 29th.

---

### Official Review · Reviewer_pmuj · 2023-07-21
**This is a dataset and benchmarks paper that ticks all the boxes: new dataset addressing existing limitations in the community, carefully constructed, publicly available, ethical implications taken into account, well designed baselines, interesting discussion on current performance and limitations of the dataset, release of raw data for posterity. A clear accept.**

**Rating:** 8
**Confidence:** 4

**Strengths:**

The main strength of this submission is the dataset itself: although there are other large room acoustics datasets in the literature, prior datasets only measure rooms with inanimate objects - whereas this dataset can be used for human tracking and identification tasks. Clearly there are several ethical implications around such a dataset, and the authors for the most part address ethical concerns well. There are other strong aspects on the construction of the dataset, for example on taking care on the microphone setup, the human locations, even the balancing of music genres for the music recordings. The fact that both raw recordings and postprocessed features are released is also a plus, as is the design of the three tasks where the dataset and baselines are evaluated on.

**Additional Feedback:**

All comments on the paper were listed in the above fields.

**Clarity:**

The paper is very well written and clear. Figures and tables are also clear and informative. There are only a couple of typos throughout the manuscript.

**Correctness:**

The dataset is very carefully constructed, and the implemented baselines are also technically correct. The evaluation methods and experimental design are also appropriate and correctly performed. Generally this is one of the strongest points of the paper.

**Documentation:**

There is a good amount of detail on data collection and organization. The dataset is publicly available under an open license, the URL is provided, and ethical concerns related to the dataset use are flagged. There is sufficient detail on the benchmarks to ensure reproducibility.

**Ethics:**

As mentioned above, I spotted a possible ethics concern regarding data collection with human participants. This possible issue might be about specific institutional policies with respect to participant consent; I understand that the dataset does not release any participant personal data, therefore institutional policies might indicate that no IRB is needed, however this would definitely need to be clarified in the manuscript. I did not identify any ethical issue regarding further use of the dataset which hasn't already been identified by the authors.

**Limitations:**

As mentioned above, the main limitation identified relates to ethical processes for participant data collection. This could already be addressed by institutional policies, however this would need to be fully clarified and justified in the paper.

**Opportunities For Improvement:**

I identified one area for improvement, which is on further discussing ethics issues with respect to data collection. Any data collection which involves human participants, whether their personal information is released or not, would typically require some level of IRB approvals. And participants would need to consent with them being recorded. This is not sufficiently discussed nor justified in the paper, where it would appear that either the authors or their institution does not deem that IRB approvals and participant consent is needed. Should the paper be accepted, I would strongly suggest that this is fully and unequivocally justified: typically any work at all involving human participants needs to go through ethical approvals and participant consent, whether or not participant personal data is released.

**Relation To Prior Work:**

The paper very clearly discusses relation to prior work with respect to large room acoustics datasets, and fully justifies the creation of this new dataset, which indeed addresses a gap in the literature on having room acoustics datasets with humans and not just inanimate objects.

**Summary And Contributions:**

This paper proposes SoundCam, a new dataset of room impulse responses and recordings in in-the-wild rooms, which can be used for various tasks related to room acoustics. The main contribution of the paper is the dataset itself, which is available under an MIT license, which addresses limitations in other room acoustics datasets. An additional contribution is the construction of baselines and experiments on related tasks, including on human localization, human identification, and human detection from audio.

---

> ### Author Response · Authors · 2023-08-23
>
> Thank you for the detailed comments about IRB approval and consent. We added a clarification in Appendix F.3 that we have followed appropriate procedures to obtain written consent from all participants for both participation in the data collection process and release of data.
>
> Regarding IRB approval,  the exemption specified in CFR 46.104 (d) (3) (i) (A) applies to our research. The exemption states that:
>
> (i) Research involving benign behavioral interventions in conjunction with the collection of information from an adult subject through verbal or written responses (including data entry) or audiovisual recording if the subject prospectively agrees to the intervention and information collection and at least one of the following criteria is met:
>
> (A) The information obtained is recorded by the investigator in such a manner that the identity of the human subjects cannot readily be ascertained, directly or through identifiers linked to the subjects;
>
> We added these details to Appendix F.3. Thank you.

---

> > ### Comment · Reviewer_pmuj · 2023-08-29
> >
> > Thank you for all the changes addressing my comments, the updates related to participant consent and IRB approval were very welcome and I will update my scores accordingly.

---

> > > ### Author Response · Authors · 2023-08-29
> > >
> > > Thank you for your comment, we were glad that your thoughtful feedback was so helpful in substantively improving our submission. Before the discussion period ends later today, we wanted to make sure there hasn't been a misunderstanding: does your current Rating reflect what you intend after our edits? Your comment mentioned changing your score, but it doesn't seem to have changed from what it was at the outset. It was nonetheless quite high in the first place, so we would fully understand if you indeed intended to maintain the same score. Thank you!

---

> > > > ### Comment · Reviewer_pmuj · 2023-08-30
> > > >
> > > > Just to confirm that the same score as before has been maintained, since as you said was high enough from the previous iteration and reflects also my current score.

---

### Official Review · Reviewer_JEJV · 2023-07-22

**Rating:** 7
**Confidence:** 4
**Correctness:** yes
**Clarity:** yes

**Strengths:**

- The dataset is comprehensive and the evaluation of baselines is extensive.
- It can be used in various task evaluations including: Locate humans using room impulse responses, Identify which human is in a room using room impulses responses, Locate humans while music is playing in the room, Determine if someone is present in the room while music is playing, Generalize localization methods to other individuals, and Test robustness of localization methods to changes in room layout.
- The generalization task for outside of the training distribution is reasonable.
- The related work is well presented and clear.

**Additional Feedback:**

See above.

**Documentation:**

yes

**Limitations:**

- I'm not an expert in the sound domain. From my understanding, the main difference between VGGish (pre-trained) and VGGish (multichannel) is 16kHz and 48kHz. So does the sample rate play a key role in the sound-related task?
- All the baselines are very simple. More advanced models like transformers and pretraining techniques, e.g., masked modeling and contrastive learning, may be explored.

**Opportunities For Improvement:**

- The diversity of humans and their poses can be explored.

**Relation To Prior Work:**

yes

**Summary And Contributions:**

The paper presents SoundCam, the largest dataset of unique RIRs from in-the-wild rooms. It includes 5,000 10-channel real-world measurements of room impulse responses and 2,000 10-channel recordings of music in three different rooms, including a controlled acoustic lab, an in-the-wild living room, and a conference room, with different humans in positions throughout each room. The dataset can be used in various sound and embodied related tasks.

---

> ### Author Response · Authors · 2023-08-23
>
> Thanks for the excellent feedback, especially pointing out the unanswered question on the importance of sample rate.
>
> **Diversity of Humans and Poses**
>
> We believe that exploring the diversity of humans and their poses is an excellent next step for extending the dataset, as real-world applications will require methods that work on many different poses, and have emphasized this in the Limitation and Conclusions section.
>
> **The Effect of Sample Rate**
>
> The sample rate is one of multiple differences between the pre-trained and multi-channel VGGish. We have edited Appendix C.2 to delineate the differences more clearly. With respect to your question, “Does the sample rate play a key role in the sound-related task?”, we conducted an experiment where we trained our multi-channel VGGish framework with audio downsampled to 16 kHz. All other variables were kept the same. Below, we compare the results between the original version and the version trained and evaluated on downsampled audio. We compare the results for localization and person identification in the treated room. These results suggest that the sample rate indeed plays a key role in the task, in that a higher sample rate is a key advantage to achieving better performance on each task. We have added and explained these results in more detail in Appendix D.2.
>
> **Effect on Localization**
> |                               | 10 mics       | 4 mics          | 2  mics        | 1  mic        |
> |-------------------------------|-------------|-------------|-------------|-------------|
> | VGGish MultiChannel (48 kHz)  | 18.1 (13.1) | 17.2 (16.2) | 20.4 (15.4) | 71.6 (50.4) |
> | VGGish  MultiChannel (16 kHz) | 18.6 (15.4) | 21.8 (16.1) | 26.8 (23.1) | 49.4 (54.3) |
>
> *The average localization error on the test set is given in centimeters, in mean (standard deviation) format.*
>
>
>
> **Effect on Identification**
> |           | 10 mics | 4 mics  | 2 mics  | 1 mic |
> |---------------------------------|----|----|----|----|
> | VGGish MultiChannel             | 82 | 81 | 76 | 64 |
> | VGGish MultiChannel Downsampled | 63 | 60 | 51 | 42 |
>
> *Classification accuracy (%) in identifying among five humans from sine sweep RIRs in the Treated Room.*
>
>
> **More Complex Baselines**
>
> The baselines are intended to be very simple, to provide several examples of ways in which the task might be addressed (traditional signal processing, deep learning). We believe they are a starting point for future improvement, and plan to try more complex approaches soon, which will likely require novel methods contributions. We invite others to join us in developing novel methods using our dataset.

---

### Official Review · Reviewer_TBmK · 2023-07-22
**Solid paper with an interesting and relevant challenge problem**

**Rating:** 7
**Confidence:** 3

**Strengths:**

The dataset appears to be well-motivated, well-constructed and rich enough to provide significant interest to the research community. The baseline results are interesting and useful, and provide a good starting point for further study.

The authors specifically highlight the potential for misuse of the techniques which could be developed using this dataset (eg. localisation of humans within private rooms using only ambient audio), and argue convincingly that making such a dataset available publicly to the community allows for further study of this area.

**Additional Feedback:**

The authors might want to consider making more documentation about the data format available on the website. In addition, a small downloadable sample dataset of a few hundred Mb might make the dataset more accessible, allowing users to quickly assess the structure of the data, especially for researchers with less internet bandwidth available.

**Clarity:**

The paper is clear and well-written. I have no concerns over the authors ability to communicate.

**Correctness:**

The dataset construction appears to be sound, and the collection regime is well documented. The baseline results are well-documented enough to be reproducible, between the paper itself and the supplementary material, however it would have been nice to have source code for the baseline experiments.

**Documentation:**

Licence information for the datasets is in the paper, but should also be made available on the website from which they are distributed (and ideally also be present in each of the downloaded files).

The authors should consider using a platform such as Zenodo to host the dataset, as it allows for clear association of a licence, as well as data versioning and the provision of a DOI for the data itself.

The authors do not provide a plan for how the dataset will be maintained, and for how long.

**Ethics:**

There is potential for this dateset to be used to develop applications which raise ethical questions, such as covert localisation and tracking of people in rooms based on ambient noise. The authors argue that release of a dataset of this sort to the community allows for open research in this domain.

The authors state that consent from the subjects was not required as the data are not personally identifiable. While I believe this is reasonable - the 3D scans are low enough resolution to prevent identification - I would be more comfortable if explicit consent had been obtained from the humans in the dataset for the recordings and 3D scans to be be released.

**Limitations:**

The authors identify that there are only a limited number of acoustic environments and human subjects present in the dataset. Obviously more variety would be better, but this is still a rich dataset as it stands. I can't see any critical points missing.

**Opportunities For Improvement:**

The scope of the dataset is primarily limited to localisation. While the authors also provide 3D scans of the human subjects and the rooms, the human identification task is limited to a small number of subjects.  The authors acknowledge this, and I don't see it as a major limitation.

**Relation To Prior Work:**

The authors review previous literature in some detail, providing a suitable amount of context for a paper of this length.

**Summary And Contributions:**

The authors present a dataset of 5000, 10-channel, room impulse responses (RIRs) for three different rooms, with humans located at different points in the room. In addition, a further 2000 recordings of music in the same rooms with humans are provided. The dataset is designed to facilitate research into both locating humans in rooms and also identifying them using audio alone.

The authors also present a set of baseline results for localisation and identification of humans in the rooms based on audio signals.

---

> ### Author Response · Authors · 2023-08-23
>
> Thank you for your valuable feedback, helping us make the dataset more accessible and ensuring it will be maintained properly.
>
>
> **Source Code for Baselines**
>
> The Source Code for the baselines will be available/maintained indefinitely at https://github.com/maswang32/soundcam
>
> **Using a Dedicated Data Hosting Platform**
>
> Thank you for this suggestion. Due to the size of our dataset exceeding the default limits of Zenodo, we elected to use Stanford Digital Repository and now have all the features you had recommended including perpetual hosting, data versioning, and a DOI. The data is now hosted here https://purl.stanford.edu/xq364hd5023.
>
> **Explicit Consent**
>
> We have added to Appendix F.3 that we obtained explicit written consent from all participants indicating their approval of the release of their data and confirming their consent to participating in the data collection process.
>
> **Plan for Maintenance**
>
> Thanks to your suggestion of using a data hosting platform, the data will be hosted indefinitely by the Stanford Digital Repository, which will track versions and maintain a persistent URL. We have provided our contact information on the hosting page for future inquiries. This information and further details have been posted on the project website and added to Appendix F.7.
>
> **Small Version of the Dataset**
>
> At your suggestion, we compiled a small subset of the dataset and included it on the website.
>
> **Information about the Dataset**
>
> We added more information about the dataset, including a schema of each of its files, on the project website.

---

> > ### Comment · Reviewer_TBmK · 2023-08-28
> > **Reviewer's Response**
> >
> > Thank you for all the changes addressing reviewer comments. The updates based on my earlier comments have significantly improved the accessibility of the data.
> >
> > One minor concern is that the Google Sites page which is the main landing page for the dataset does not yet appear to link to the Stanford Digital Repository version of the dataset but still provides the original download links. This should be addressed.

---

> > > ### Author Response · Authors · 2023-08-29
> > >
> > > Thank you for pointing this out. We have updated the website to reference the dataset on the Stanford Digital Repository.

---

### Official Review · Reviewer_m2YB · 2023-07-22
**Limited use case**

**Rating:** 5
**Confidence:** 3

**Strengths:**

- The paper is clearly written.
- Authors address the limitation of the tested models.
- Tasks themselves are academically interesting.

**Additional Feedback:**

I respect the authors efforts on data collection and investigation on the sound-based human detection/localization/identification. They sound indeed academically interesting. However, I do not find any strong motivation to invite the community to improve performance on these tasks as the generalization capability seems to be inherently limited and there are better approaches for human detection/localization/identification while addressing privacy.

**Clarity:**

- Fairly clear.

**Correctness:**

- Probably correct.

**Documentation:**

- RT60 for each room should be reported.

**Ethics:**

As images and voices are not included, I have no concerns.

**Limitations:**

It is highly doubtful that the proposed sound-based human detection, localization, identification will be applied to real-world. Although authors mention about the privacy issue of camera-based approaches, sound recording with microphones also raise privacy issues. Other sensors such as IR camera may provide more reliable results while addressing the privacy issues.
Also, all methods/tasks are room specific, meaning that it requires expensive data collection for each room to train a model. Even during the inference time, it needs to playback sweep sound or known time-synchronized music, which may not be preferable. Even in the acoustically treated room, models already are shown to struggle in generalization.

**Opportunities For Improvement:**

RT60 for each room should be provided.

**Relation To Prior Work:**

- Although prior works are fairly introduced, the motivation of RIR-based approaches is week.

**Summary And Contributions:**

The dataset of 5,000 10-channel real-world measurements of room impulse responses (RIRs) and 2,000 10-channel recordings of music in three different rooms. The main motivation of this dataset is to build a system that localize, identify, or detect a human in the room from sounds.
Simple systems are tested in the following tasks:
- Human Localization: Estimating the location of human in a known room on the basis of sweep-based or music-based RIR,
- Human Identification: Identifying a human identity of 5 known persons in a known (semi-anechoic) room from sweep-based RIR,
- Human Detection: Detecting the existence of human in a known (semi-anechoic) room from music-based RIR or music recording.

While experimental results show that it is possible to address these tasks using deep neural networks, it also shows the limited generalization capability.

---

> ### Author Response · Authors · 2023-08-23
>
> Thank you for your feedback, especially your detailed comments about the usefulness of these tasks. We agree that there are inevitably tradeoffs with privacy of audio versus vision or infrared signals, as any signal that can be used to perform even rudimentary localization or identification will have some inherent privacy concerns.  We claim that the current limitations of sound-based approaches are otherwise impractical to benchmark and mitigate without a robust, properly collected dataset like SoundCam. We therefore hope that our dataset and benchmark tasks motivate future methods contributions which significantly improve upon these baselines in terms of generalization, sample efficiency, etc., leading to more viable sound-based localization methods.
>
> **Regarding Other Methods**
>
> While IR cameras may provide more reliable results at the moment, we believe our sound-based tasks and approaches are useful to explore. Microphones are ubiquitous and already exist in many consumer devices, so they avoid the cost of adding significant additional hardware, like cameras. Microphones are cheap, and infrared cameras are expensive and could require additional installations. Also, an omnidirectional microphone can work when the device is in any orientation, and does not require it to be oriented in a specific direction.
>
> Also, since sound waves diffract around walls and obstacles, it is possible that sound-based localization techniques can provide information complementary to vision or infrared signals, neither of which are robust to occlusion.
>
> **Cost of Data Collection**
>
> In each dataset, we collect 1,000 RIRs for the purpose of providing a comprehensive and multipurpose dataset. Collecting this many points takes about 10 hours. However, SoundCam can be used to develop methods that require significantly fewer data points. In fact, one of our baseline methods, the Time of Arrival Method, is an analytical method requiring no training data at all, only measurements of microphone positions and a measurement of the empty room’s impulse response.
>
> Below, we also show results of testing sample efficiency for the multi-channel VGGish on the localization task, using all microphones, in the real living room. With just 100 data points, we are able to localize the human to within 68 cm.
>
> | Training Set Size | Mean Error (std) |
> |-------------------|------------------|
> | 25                | 166.5 (61.1)     |
> | 50                | 112.9 (78.3)     |
> | 100               | 67.0 (51.1)      |
> | 200               | 52.2 (37.9)      |
> | 500               | 31.5 (24.1)      |
> | 800               | 27.9 (22.0)      |
>
> *The average localization error on the test set is given in cm, in mean (standard deviation) format.*
>
>
> **Inference**
>
> For each of our tasks, inference requires playing sound. We have added to the Limitations and Conclusions section that we welcome future methods contributions which can ignore the ground truth synchronized source signals we have included, and instead perform these benchmark tasks merely from the recordings we have provided, e.g. through blind deconvolution or some other technique.
>
> **Generalization Capability**
>
>  We see two types of generalization which are each important for localization techniques: generalizing across changes of configuration of the same room and generalizing across completely different rooms. Our results from attempting to generalize across our different configurations of the treated room confirm that generalizing to new room configurations is already quite difficult, and generalizing to a new room is ill-defined for our current baselines. However, our dataset can be used for benchmarking a future method on each of these types of generalization. We have made Section 4.1 clearer in this regard.
>
> **Prior Work and Motivation for RIR-based Approaches**
>
> While there is no perfect way of characterizing an environment’s acoustic field, RIRs are perhaps the most standard way of doing so. In addition to [Yang et al., 2022], we have also added [Antonacci et al., 2012], [Aprea et al., 2009], and [Tervo et al., 2012] to our Related Work.
>
> **RT60s**
>
> Below are the RT60s for each room/configuration, which will be provided along with the dataset.
>
> | Room                   | Average RT60 (s) | Min RT60 (s) | Max RT60 (s) |
> |------------------------|------------------|--------------|--------------|
> | Treated Room           | 0.158            | 0.112        | 0.264        |
> | Treated Room w/ Panels | 0.158            | 0.111        | 0.248        |
> | Living Room            | 1.121            | 1.022        | 1.170        |
> | Conference Room        | 0.581            | 0.541        | 0.608        |
>
> *RT60s in each room/configuration. The first column shows the average RT60 across all microphones, while the second and third columns show the minimum and maximum across all microphones, respectively.*
>
> We have added this table and detailed graphs of RT60 by frequency band for each room in Appendix B.4.

---

> ### Author Response · Authors · 2023-08-28
>
> Thank you again for your valuable feedback. We hope our edits addressed your comments. Please let us know if there are any additional clarifications or edits you would advise. We would appreciate any additional feedback to further improve our draft as much as we can before the end of the rebuttal period on the 29th.

---

> ### Author Response · Authors · 2023-08-29
>
> Thank you again for your thoughtful feedback. We hope that the edits we have made reflect that we understood your concerns and have addressed them. We would be very grateful if you could let us know whether or not our edits have addressed your concerns and/or changed your opinion on the quality of the paper before the rebuttal period ends today.

---

### Author Response · Authors · 2023-08-23
**Summary of Responses**

Thank you to all our reviewers for the valuable feedback which helped us improve the rigor of our experiments, the accessibility of our dataset, and the clarity of our writing. Though we have posted individual responses to each reviewer’s feedback, the following is a summary of the main points from the reviews as well as the main changes we have made at our reviewer’s suggestions.

**Main Requests from Reviewers**
- Obtain explicit consent from participants in the data collection process
- Additional information regarding room reverberation times, silence levels and content, and deconvolution methods
- Investigation on the effect of sample rate on baseline performance
- Verify correctness of baselines
- Additional information regarding dataset accessibility, hosting, and maintenance
- Writing clarifications



**Summary of Changes**
- Written consent from participants in the data collection process
- Additional measurements of room reverberation time and content and levels of silence recordings
- Additional results regarding data ablation and the effect of audio sample rate on baseline performance
- Adjustment to the binary detection baseline
- Migration to dedicated data hosting platform and additional information on the website about data organization, as well as a sample dataset
- Clarifying edits

---

### Decision · Program_Chairs · 2023-09-22

**Decision:**

Accept (Poster)

**Comment:**

On balance I recommend acceptance of this submission. The authors address an important issue that is of wide interest to researchers, the public and industry. Issues around ethical concerns are somewhat mitigated by the fact that this brings this type of research into the open.
The genie is out of the bottle with in-home acoustic surveillance.

quality: A high quality article, well structured and well evaluated. There are some issues around active localization vs passive, which could be addressed in the next submission.
clarity: generally well described, except for minor points raised by reviewers.
originality: There are other acoustic localization databases out there, but this adds to the general field, in both terms of data and analysis.
significance: The wide use of voice-activated systems means that a microphone-based tracking a system could be used for many purposes, from health to intruder alarms. The introduction of a database that enables acoustic presence, location and identity tracking, allows for the augmentation of existing devices. Although other sensors might be better suited to the task, this doesn't detract from the importance, since it could be a way to lower costs or improve tracking of other sensors.